# p53 isoforms have a high aggregation propensity, interact with chaperones and lack binding to p53 interaction partners

Anamari Brdar[1†], Christian Osterburg[1*†], Philipp Münick[1†], Anne Christin Machel[1], Rajeshwari Rathore[2], Susanne Osterburg[1], Büşra Yüksel[1,3], Birgit Schäfer[1], Kristina Desch[4,5], Julian D Langer[5], Ivan Dikic[2,6,7], Volker Dötsch[1*]

[1]Institute of Biophysical Chemistry and Center for Biomolecular Magnetic Resonance, Goethe University, Frankfurt am Main, Germany; [2]Institute of Biochemistry II, Faculty of Medicine, Goethe University Frankfurt, Frankfurt am Main, Germany; [3]IMPRS on Cellular Biophysics, Frankfurt, Germany; [4]Max Planck Institute for Brain Research, Frankfurt, Germany; [5]Max Planck Institute of Biophysics, Frankfurt, Germany; [6]Buchmann Institute for Molecular Life Sciences, Goethe University Frankfurt, Frankfurt am Main, Germany; [7]Fraunhofer Institute of Translational Medicine and Pharmacology, Frankfurt am Main, Germany

*For correspondence:
c_osterburg@mailbox.org (CO);
vdoetsch@em.uni-frankfurt.de
(VD)

†These authors contributed
equally to this work

## eLife Assessment

This manuscript provides an **important** biochemical analysis of p53 isoforms, highlighting their aggregation propensity, interaction with chaperones, and dominant-negative effects on p53 family members. The authors have substantially strengthened the original manuscript by incorporating new mass spectrometry data and clarifying isoform-specific oligomerization behavior. Although the use of high expression levels limits direct physiological interpretation, the work is carefully framed as an investigation of protein misfolding and stability. Overall, this study offers **convincing** insights into p53 isoform biophysics with broad implications for cancer biology.

**Abstract** The p53 transcription factor family consists of the three members p53, p63, and p73. Both p63 and p73 exist in different isoforms that are well characterized. Isoforms have also been identified for p53 and it has been proposed that they are responsible for increased cancer metastasis. In contrast to the p63 and p73 isoforms, which do not contain truncations in folded domains, most of the p53 isoforms contain only parts of either the DNA-binding domain (DBD) or the oligomerization domain. To better understand the effect of p53 isoforms in cancer, we provide here a comprehensive biochemical characterization. With the exception of the Δ40p53α isoform, none of the other variants can bind to DNA with high affinity and none can upregulate transcription. Probing with antibodies, DARPins and other interaction partners confirmed that isoforms harbouring deletions in the DBD cannot interact specifically with them, but instead are bound to chaperones and other factors known to interact with misfolded proteins. Expression of isoforms with deletions in the DBD results in upregulation of cellular chaperones. If the expression level surpasses a threshold, the chaperone system can no longer keep these isoforms soluble, resulting in aggregation and co-aggregation with other factors.

## Introduction

The tumour suppressor p53 is the most mutated protein in cancer cells with more than 2000 different mutations identified so far (*Joerger et al., 2006*; *Funk et al., 2025*). In tumour cells with wild-type p53, its activity is also inhibited for example by the overexpression of E3 ligases such as Mdm2 (*Haupt et al., 1997*; *Kubbutat et al., 1997*). p53 is a transcription factor that sequence-specifically binds to DNA to regulate the transcription of genes involved in apoptosis and cell cycle inhibition as the two most direct processes that prevent tumourigenesis. In addition, p53 also regulates genes involved in many other processes such as DNA repair and metabolism. Structurally, it consists of a central DNA-binding domain (DBD), an N-terminal transactivation domain (TAD) and a C-terminal oligomerization domain (OD) through which it forms tetramers (*Joerger and Fersht, 2007*). At the very C-terminus, an additional regulatory domain (CTD) is located that binds unspecifically to DNA and contains multiple sites for phosphorylation, ubiquitination, and other post-translational modifications. Mutations can occur in all these domains but mainly cluster in the DBD (*Joerger et al., 2006*; *Funk et al., 2025*). High-affinity DNA binding requires the formation of tetramers which enables dominant negative effects of cancer mutations in the DBD as mutant p53 and wild-type p53 can form mixed tetramers via their ODs with strongly reduced binding affinity. Similar dominant negative effects are known from the p53 family member p63 in which mutations in the DBD result in developmental defects based on a dominant negative mechanism (*Osterburg et al., 2023*; *Celli et al., 1999*). The discovery of p63 and the third family member p73 has revealed that these two proteins exist in multiple different isoforms that are created by the combination of different promoters with splicing events in the C-terminus which strongly influences the transcriptional activity of these isoforms (*Osterburg and Dötsch, 2022*). Different isoforms have also been identified for p53 (*Bourdon et al., 2005*) and it has been proposed that these isoforms are responsible for increased cancer metastasis as well as premature aging upon dysregulation. In contrast to the p63 and p73 isoforms, which do not contain truncations in folded domains, most of the isoforms of p53 contain only parts of either the DBD or the OD. All these deletions have strong effects on the biochemical behaviour of the isoforms (*Bourdon et al., 2005*; *Graupner et al., 2009*). A deletion of the OD not only prevents the formation of tetramers which are required for high-affinity DNA binding, it also deletes the nuclear export signal, thereby influencing the subcellular localization of these isoforms. The wild-type p53 DBD is known to be only metastable with a melting temperature of ~43°C (*Joerger et al., 2006*; *Bullock et al., 1997*). Many p53 mutations found in tumours destabilize the DBD further, resulting in unfolding and due to the exposure of aggregation-prone peptide sequences in gain of function effects via co-aggregation with other cellular factors (*Xu et al., 2011*; *Levine et al., 1995*). This metastability, combined with the in general cooperative folding behaviour of single domain proteins, strongly predicts that the isoforms containing a truncated DBD cannot properly fold but have a high aggregation potential, similar to mutant p53 with destabilizing mutations.

p53 isoforms have been studied in different cellular systems and individual isoforms have been assigned specific regulatory functions (*Zhang et al., 2019*; *Horikawa et al., 2014*; *Gadea et al., 2016*; *Roth et al., 2016*; *Fujita et al., 2009*; *Mondal et al., 2013*; *Turnquist et al., 2016*; *von Muhlinen et al., 2018*; *Gong et al., 2015*). However, a comprehensive biochemical characterization of these isoforms is missing and we propose that many of the observed effects can be assigned to co-aggregation of these isoforms with wild-type p53 as well as other cellular factors such as the other members of the p53 protein family. Here, we report a comprehensive characterization of the p53 isoforms with respect to their stability, DNA-binding affinity, interaction with chaperones, aggregation and co-aggregation.

## Results

### All isoforms except wtp53 show a low to no transcriptional activity

So far, a total of 14 p53 isoforms with various truncations have been identified. Compared to the wild-type p53 form (from now on named wtp53) alternative translation start sites create the isoforms Δ40p53, Δ133p53, and Δ160p53, each lacking the indicated number of amino acids from the N-terminus (*Joruiz and Bourdon, 2016*). The Δ40p53 variants have a deletion of the first 40 amino acids of the N-terminus including the transactivation domain 1 (TAD1) (*Krois et al., 2016*) but contain a wild-type DBD and OD. The Δ133p53 and Δ160p53 isoforms lack, in addition to the complete TAD,

parts of the DBD. These different N-terminal isoforms can be combined with three different C-termini called α, β, and γ. Only the wild-type αC-terminus contains a full OD, the β- and γC-termini have a truncated OD with isoform-specific extensions. Furthermore, two isoforms with C-terminal deletions of parts of the DBD (TAΔp53α) (*Rohaly et al., 2005*) and additional deletion of the remaining C-terminus including the entire OD (TAp53ψ) (*Senturk et al., 2014*) have been described. *Figure 1* shows an overview of all these isoforms.

As an initial experiment, we determined the transcriptional activity of the p53 isoforms in relation to wtp53 and different classes of p53 mutants both in H1299 and in SAOS-2 cells using reporter plasmids containing either the multiple repeats of a p53 RE (pBDS), p21, or Mdm2 promoters. All three promoters are very sensitive to p53. Hence, p53 isoforms which show weak or no response in luciferase assays probably will not show any activity under endogenous protein expression levels on genomic promoters. To exclude the possibility that lower p53 isoform activities are due to reduced expression levels, a titration series with increasing amounts of wtp53 plasmid DNA was measured for comparison. In addition, we investigated artificial constructs that contained various combinations of full-length domains (DBD-OD-CTD; TA-DBD-OD; TA-DBD). wt-p53 showed a strong transcriptional activity (*Figure 2A–C*, *Figure 2—figure supplement 1*). An even higher activity was seen for the artificial construct TA-DBD-OD, lacking the CTD. As a negative control, we used the p53R175H mutant, which harbours a structural DBD mutation leading to a complete loss of transcriptional activity. TAp53β and TAp53γ still contain the full-length TA and DBD domains but lack the OD and therefore are unable to form tetramers. They showed a slight transactivation capacity, which is, however, several-fold lower than the wtp53 activity. This slight capacity could be due to weak binding of the p53 monomers to DNA and their intact TA domain, since isolated p53 DBDs can still cooperatively bind DNA with low affinity. This result is also consistent with the activity seen for the artificial TA-DBD construct. In agreement with a previous study (*Chan and Poon, 2007*), no transactivation capacity was detected for TAΔp53α which lacks part of its DBD and linker region to its OD as well as the nuclear localization signal (NLS) and for the p53ψ isoform (*Figure 2D*, *Figure 2—figure supplement 1*). Δ40p53α and Δ40p53β showed a slight transactivation capacity (100- to 200-fold below wt-p53). Δ40p53α lacks part of its TAD but preserves its OD while Δ40p53β lacks part of its TAD as well as of its OD domains. The artificial construct DBD-OD-CTD had a lower activity, as it lacks the TAD completely, while the TAD2 part is still present in the Δ40 isoforms (*Osterburg and Dötsch, 2022*; *Krois et al., 2016*). The other N-terminal truncations (Δ133 and Δ160) showed no activity at all, independent of their C-terminus (α, β, and γ), consistent with a lack of the entire TAD and parts of the DBD as well as depending on the C-terminus a lack of the OD (*Figure 2A–C*).

The expression levels of the constructs differed with β-isoforms being much stronger expressed compared to α-isoforms (*Figure 2—figure supplement 1*). The γ-isoforms showed the lowest expression of all C-terminal truncations, while the lowest expression levels among the N-terminal isoforms were seen for the Δ133p53 isoforms. These data also demonstrated that the lowest wtp53 concentration had a much higher transcriptional activity compared to all other p53 isoforms, even when these isoforms reached concentrations many times above the concentration of wtp53. Taken together, some p53 isoforms showed a higher transactivation capacity compared to others, but no isoform reached the level of wtp53. Moreover, the titration series with wtp53 indicated that saturation in this assay is achieved above transfection with 50 ng plasmid. As we had used 267 ng for all isoforms, any remaining transcriptional activity should have been detected in this assay.

## Only wtp53 and Δ40p53α bind DNA in pulldown assays

Next, we investigated the ability of the various p53 isoforms to bind to DNA using a pulldown experiment with the response elements (REs) of the p21 and the PUMA promoter. Only wtp53 and Δ40p53α showed binding to the p53-REs (*Figure 2E*, *Figure 2—figure supplement 2*). For high-affinity DNA binding, p53 family members need a folded DBD and the ability to form tetramers. This requirement is met only by wtp53 and Δ40p53α. As a negative control, we used again the p53αR175H mutant which is unable to bind DNA due to its destabilized DBD. For all other p53 isoforms, no DNA binding could be observed due to the lack of a folded DBD (Δ133, Δ160 constructs, and p53ψ), lack of a functional OD (β- and γ-C-termini) or a combination of both. We confirmed the results of these pulldown assays for Δ40p53β by surface plasmon resonance (SPR) measurements with the p21 RE being immobilized on the chip (*Figure 2F*, *Figure 2—figure supplement 2*). The artificial DBD-OD construct showed

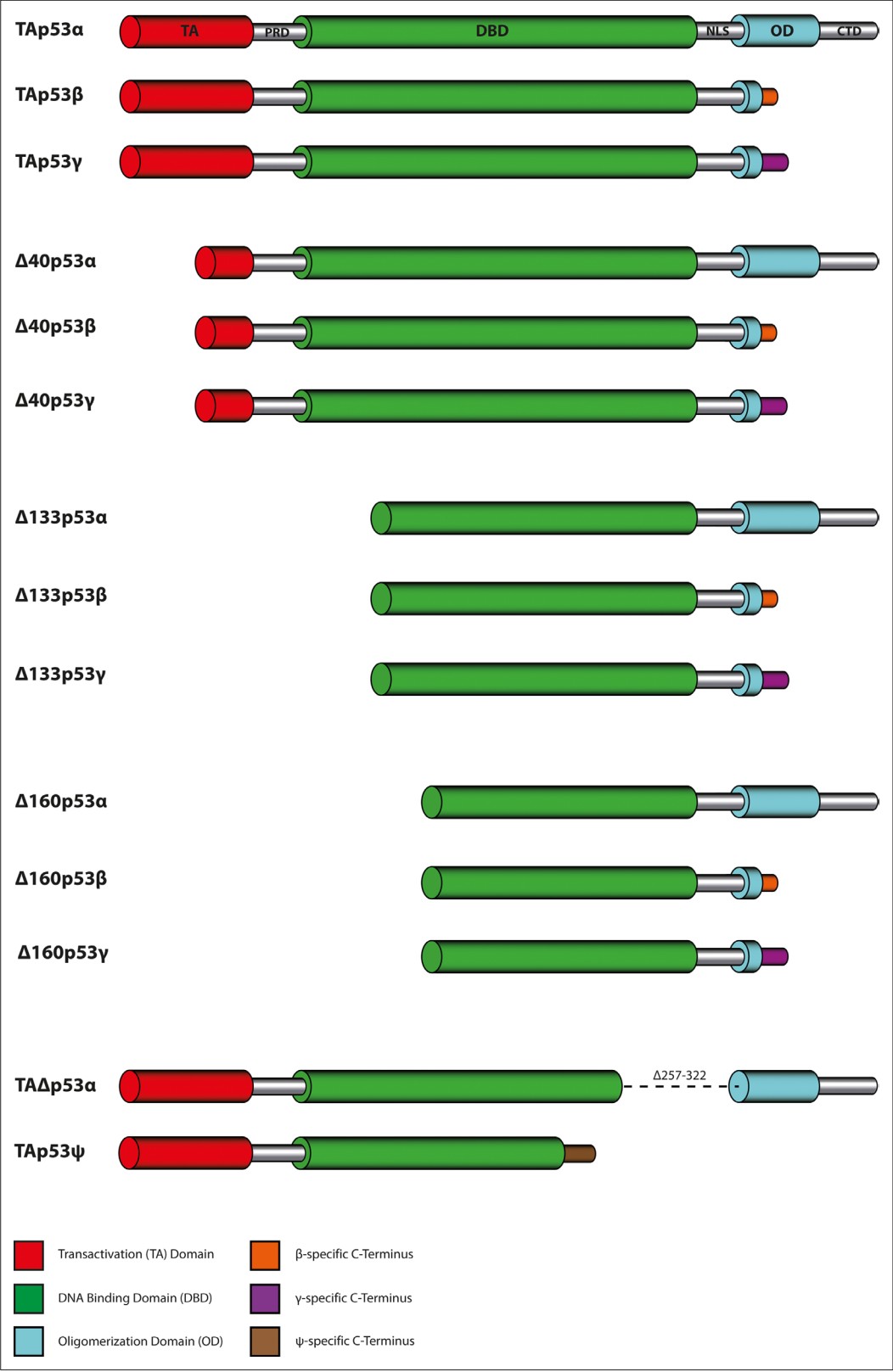

**Figure 1.** Domain organization of p53 isoforms. Overview of the so far described p53 isoforms that are formed by a combination of four N-terminal (TAp53, Δ40p53, Δ133p53, and Δ160p53) and three C-terminal variants (α, β, and γ), leading to 12 individual p53 isoforms. The remaining two isoforms p53$\phi$ (or TAp53$\phi$) and Δp53 (or TAΔp53α) are generated by alternative splicing.

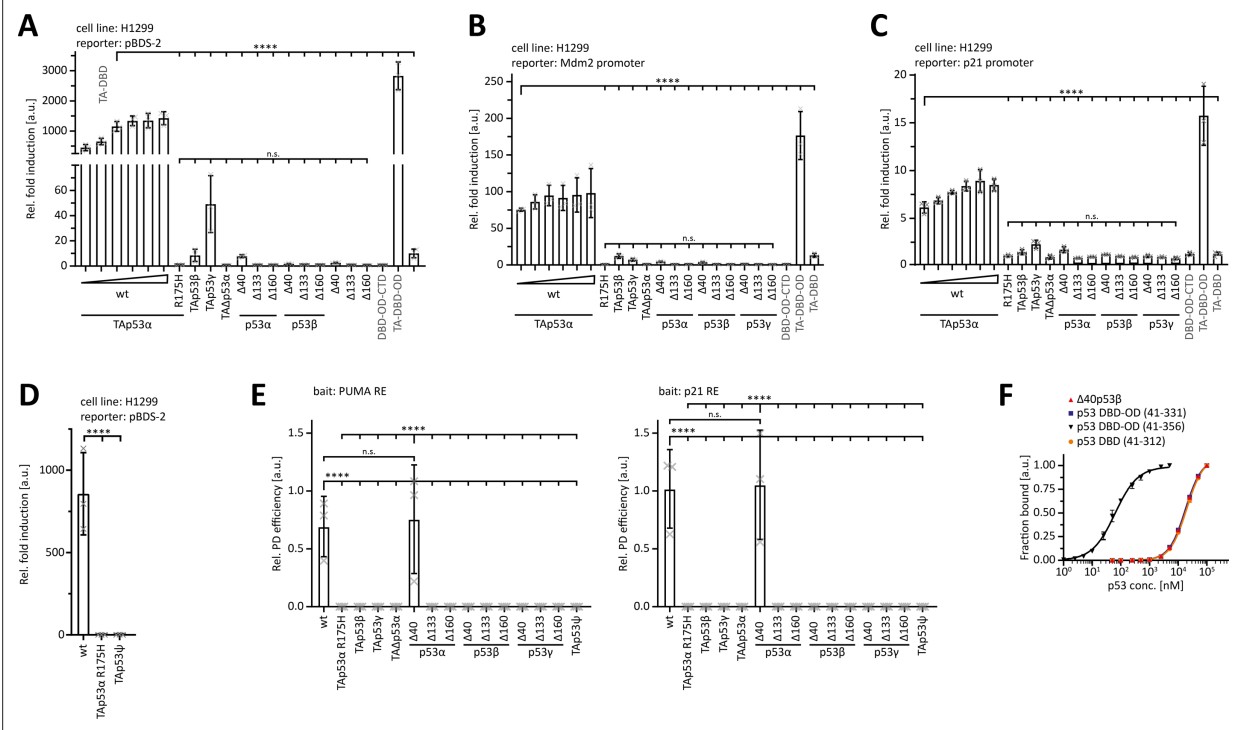

**Figure 2.** Activity, oligomerization, and DNA binding. Luciferase reporter assay of p53 isoforms and variants either on the pBDS-2 reporter (three repeats of the 14-3-3σ promoter RE) (**A**), or on the Mdm2 promoter (**B**), or on the p21 promoter (**C**). TAp53α carrying the cancer-related R175H mutation served as a negative control. H1299 cells (**A, B**) or Saos-2 cells (**C**) were transiently transfected with the respective luciferase reporter plasmids and the N-terminally Myc-tagged proteins. The plasmid encoding for TAp53α was titrated with the remaining DNA amount for transfection being filled up with empty vector. (**D**) Luciferase reporter assay of p53 $\phi$ on the pBDS-2 reporter. TAp53α carrying the cancer-related R175H mutation served as a negative control. H1299 cells were transiently transfected with the respective luciferase reporter plasmids and the N-terminally Myc-tagged proteins. (**E**) DNA pulldown assay of p53 isoforms with the 20 bp REs of the human PUMA and p21 promoter as bait. TAp53α carrying the cancer-related R175H mutation served as a negative control. N-terminally Myc-tagged proteins were in vitro translated using rabbit reticulocyte lysate (RRL). For the relative pulldown efficiency, each pulldown signal was normalized to the input signal. (**F**) Surface plasmon resonance (SPR) affinity curves of purified Δ40p53β (red), p53 DBD-OD variants (purple and black), and the isolated p53 DBD (orange) binding to the 20bp p21 response element (RE) immobilized on a streptavidin (SA) chip. Data points were extracted by equilibrium analysis of sensograms as in *Figure 2—figure supplement 2*, plotted and fitted with a non-linear, least squares regression using a single-exponential one-site binding model with Hill slope. p53 DBD-OD (41356) encompasses the complete OD, while p53 DBD-OD (41–331) only has the N-terminal part of the OD common to all p53α, p53β, and p53γ isoforms. (**A–E**) The bar diagram shows the mean values and error bars the corresponding SD ($n = 3$). Statistical significance was assessed by ordinary one-way ANOVA (n.s.: $p > 0.05$, $*p \leq 0.05$, $**p \leq 0.01$, $***p \leq 0.001$, $****p \leq 0.0001$).

The online version of this article includes the following source data and figure supplement(s) for figure 2:

**Figure supplement 1.** Activity, of p53 isoforms.

**Figure supplement 1—source data 1.** Uncropped Western blots.

**Figure supplement 2.** DNA binding of p53 isoforms.

**Figure supplement 2—source data 1.** Uncropped Western blots.

high affinity, while the affinity for the Δ40p53β was ~280-fold lower and comparable to the affinity of the isolated p53 DBD (*Figure 2—figure supplement 2*). For all other p53 isoforms, SPR measurements were not feasible because these proteins could not be purified following bacterial expression. Summarizing the transactivation and DNA-binding assays, all p53 isoforms show a behaviour more comparable to mutant p53 than to wtp53. Only Δ40p53α is able to bind DNA without exhibiting a significant transcriptional activity.

## Only wtp53 and Δ40p53 isoforms have an intact DBD

We further investigated the integrity of the DBD with binding to different interaction partners – the Human Papilloma Virus (HPV) E6 protein, a recently described DARPin and the pAB240 antibody – that probe different interfaces. The E6 protein binds to a region of the p53 DBD different from the

DNA-binding interface (*Figure 3A*) and initiates the degradation of p53 by the ubiquitin proteasome system (*Martinez-Zapien et al., 2016*; *Wang et al., 2024*; *Münick et al., 2025*). Degradation requires the recognition of the p53 DBD by the E6 protein. An unfolded DBD or a DBD with a non-wild-type conformation does not get degraded. We expressed various p53 isoforms in rabbit reticulate lysate and determined the levels of the isoforms 4 hr after the addition of bacterially expressed GST-E6 protein or GST as control. As expected, wtp53 was completely degraded in this assay (*Figure 3B*, *Figure 3—figure supplement 1*). The structural mutant p53R175H was only partially degraded. This mutation locally perturbs the structure of the DBD, but also reduces the overall thermodynamic stability. Another structural mutant V157F was not degraded at all. The R273H mutation that is located in the DNA-binding interface did not inhibit degradation, demonstrating that it retains the wild-type fold of the DBD. TAp53ϕ and TAΔp53α showed no decreased signal in the presence of HPV E6. Both lack C-terminal parts of the DBD, indicating that these isoforms are not able to fold their DBD correctly.

Isoforms that contain a complete DBD should be sensitive to this degradation assay and consequently, TAp53β, TAp53γ as well as Δ40p53β and Δ40p53γ are degraded, although not as efficiently as wtp53. This is consistent with a similar result for the artificial construct TA-DBD. A less efficient degradation might not directly reflect differences in the binding efficiency of the E6 protein but also differences in the accessibility of lysine residues necessary for ubiquitination. This effect can be seen in the less efficient degradation of the TAp53αΔCTD construct in which the lysine-rich CTD is missing (*Figure 3B*). The remaining p53 isoforms with deletions in the DBD (Δ133p53α-γ and Δ160p53α-γ) show only low levels or even no degradation, similar to ΔNp63α which we have added as a negative control. While the explanation for the insensitivity of the Δ133p53 and Δ160p53 isoforms is that the missing parts of the DBD include part of the binding interface of the E6 protein, this interface is in principle present in the TAp53ϕ and TAΔp53α isoforms. Their resistance to degradation shows that the remaining DBD does not adopt a fold that can be recognized by the E6 protein.

As a further experiment to investigate the conformational state of the DBD, we probed isoforms expressed in H1299 cells by immunoprecipitation with the pAB240 antibody (*Figure 3C*, *Figure 3—figure supplement 2*). This antibody detects an epitope in the p53 DBD that becomes accessible only upon unfolding of the domain. The results showed that the two p53 mutants R175H and V157F were efficiently immunoprecipitated (IP). Likewise, all Δ133 and in particular all Δ160 isoforms as well as p53ϕ and TAΔp53α interacted with this antibody.

Finally, to investigate the structural integrity of the DBD, we performed binding studies with the DARPin G4 which we had previously shown to bind to the DNA-binding interface of p53 (*Strubel et al., 2022*). For pulldown experiments, we bound the biotinylated DARPin to streptavidin beads and expressed different p53 isoforms and mutants either in rabbit reticulate lysate (*Figure 3D*, *Figure 3—figure supplement 2*) or in H1299 cells (*Figure 3E*, *Figure 3—figure supplement 2*) (Δ40p53 and Δ160p53 isoforms were not further tested as the first ones contain a fully folded DBD and results with Δ133p53 are directly transferable to Δ160p53). The G4 DARPin pulled down wtp53 as well as the R273H mutant. The R175H mutant was also pulled down, albeit with a lower efficiency with H1299 cells expressed protein (due to the higher temperature of cell culture compared to expression in rabbit reticulocyte lysate (RRL) which results in a lower level of the DBD with wild-type conformation). The efficiency for the V157F mutant was very low. None of the Δ133 isoforms or the p53ϕ and TAΔp53α showed interaction.

Together, these binding studies demonstrated that only wtp53 and the Δ40p53 variants contain a fully folded DBD. High-affinity DNA binding in addition requires a functional OD which enables only wtp53 and Δ40p53α to interact strongly.

## The γ-C-terminus and isoforms with truncated DBDs exhibit a high aggregation propensity in silico, in vitro, and in cellulo

The only partial folding of many of the p53 isoforms triggered the question whether the unfolded state of the DBD and OD leads to changes in the solubility within the cell. The DBD contains an aggregation-prone peptide sequence that is usually hidden within the folded structure but gets exposed upon unfolding (*Wang and Fersht, 2015*). In the case of p63, we had previously shown that mutations within the SAM domain also led to the exposure of a usually hidden aggregation-prone peptide which results in aggregation of p63 (*Russo et al., 2018*) as the cause of the ankyloblepharon-ectodermal

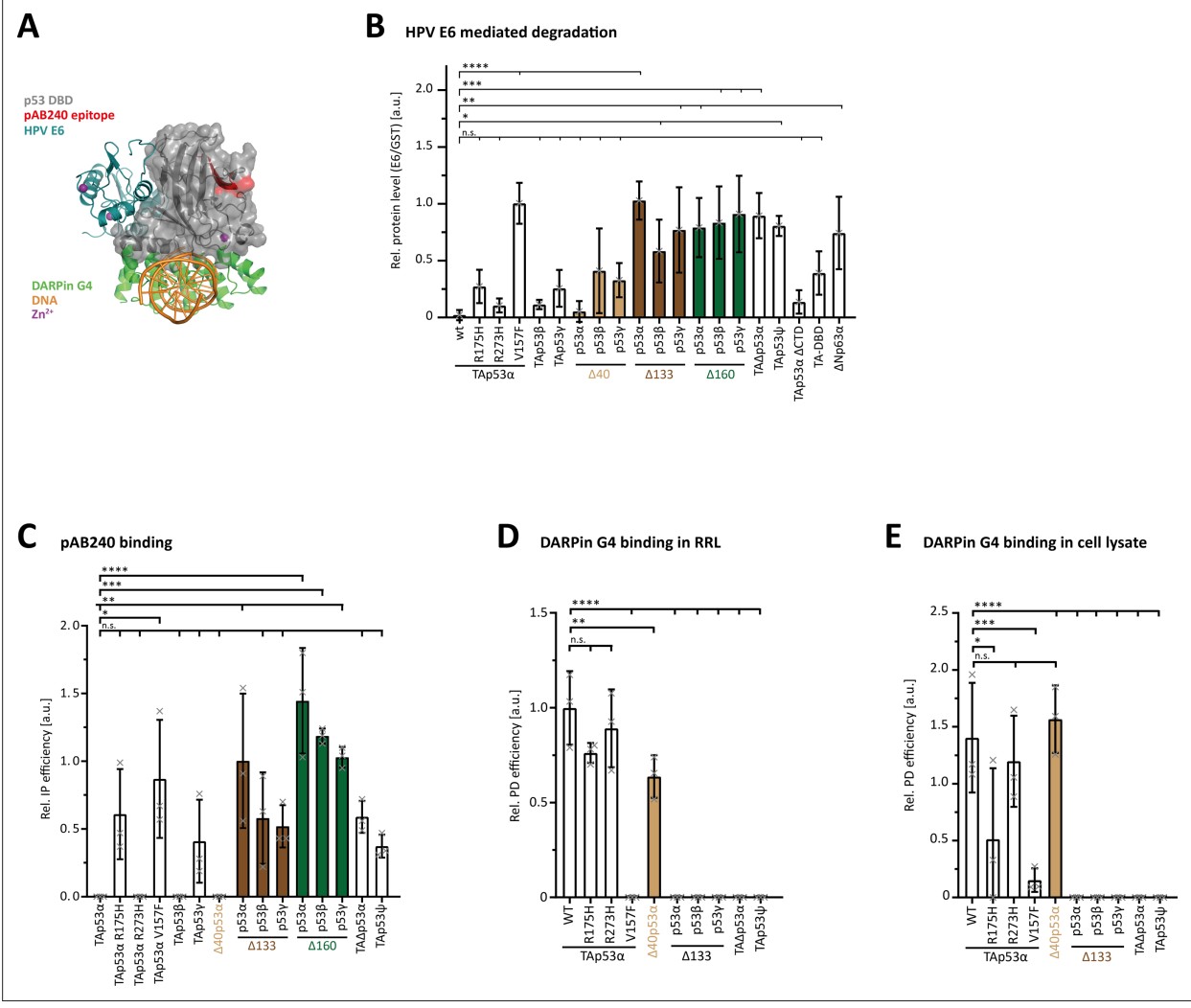

**Figure 3.** Probing the DNA-binding domain (DBD) fold. (**A**) Superposition of structures of the p53 DBD (grey) in complex with DNA (orange; PDB: 3TS8), HPV16 E6 (blue; PDB: 4XR8), and DARPin G4 (green; PDB: 7Z7E). Epitope of the pAB240 is marked in red and the zinc ion is depicted as a purple sphere. (**B**) E6 degradation assay of p53 isoforms, cancer-related mutants, and variants. ΔNp63α served as a negative control. N-terminally Myc-tagged proteins were in vitro translated using rabbit reticulocyte lysate (RRL). Lysates were diluted in reaction buffer and supplemented with either 5 µM GST-tagged HPV16 E6 or GST only as control. Reactions were incubated for 4 hr at 25°C and analysed for protein levels by WB. Signals were normalized to the loading control vinculin and the relative protein level ratio between E6 and GST samples was calculated by setting the normalized signal of the GST samples to 1. (**C**) Conformation-specific immunoprecipitation (Conf-IP) of p53 isoforms and cancer-related mutants. H1299 cells were transiently transfected with empty vector or N-terminally Myc-tagged p53 variants. p53 was immunoprecipitated (IP) with either α-mouse IgG or an α-p53 antibody (pAB240). The latter binds an epitope in the DBD of p53, which is only exposed when the domain is unfolded. Consequently, pAB240 only recognizes intrinsically unfolded p53 mutants under native IP conditions. Input and IP samples were subsequently analysed by WB using αMyc antibody. For the relative Conf-IP efficiency, each IP signal was normalized to the input signal. (**D**) DARPin pulldown assay of p53 isoforms and cancer-related mutants in vitro translated using RRL or transiently expressed in H1299 (**E**) with immobilized DARPin G4. DARPin G4 only recognizes the folded DBD of p53. For the relative pulldown efficiency, each pulldown signal was normalized to the input signal. Note that the expression temperature in the RRL experiment was 30°C lower than the temperature in the cell culture experiment (37°C), resulting in a higher level of folded R175H p53 mutant and hence a higher pulldown efficiency. (**B–E**) The bar diagram shows the mean values and error bars the corresponding SD (*n* = 3). Statistical significance was assessed by ordinary one-way ANOVA (n.s.: p > 0.05, *p ≤ 0.05, **p ≤ 0.01, ***p ≤ 0.001, ****p ≤ 0.0001).

The online version of this article includes the following source data and figure supplement(s) for figure 3:

**Figure supplement 1.** Probing the DBD Fold by degradation via interaction with E6 protein.

**Figure supplement 1—source data 1.** Uncropped Western blots.

**Figure supplement 2.** Co-IP experiments with different binding partners.

**Figure supplement 2—source data 1.** Uncropped Western blots.

defects-cleft lip/palate (AEC) syndrome, in human patients (**McGrath et al., 2001**). In addition to point mutations in the SAM domain, frame shift mutations also cause the AEC syndrome by creating new aggregation-prone sequences (**Russo et al., 2018**). We investigated the potential for such newly formed aggregation-prone regions (APRs) in the sequences of the β- and γ- as well as the TAp53φ isoforms by an in silico prediction method using the TANGO algorithm (**Fernandez-Escamilla et al., 2004**). This analysis predicted indeed the presence of novel APRs in the DBD of TAp53φ and the OD of p53γ (**Figure 4A–C**).

Experimentally, we investigated the aggregation behaviour of H1299 expressed p53 isoforms by analytical size-exclusion chromatography (SEC). The results indicated a tetrameric state for wtp53 as well as for p53R273H and Δ40p53α (**Figure 4D**). The cancer mutant p53R175H showed a mixture of tetrameric protein and partial aggregation indicated by a shift to higher molecular weight. TAp53β and Δ40p53β showed a shift towards lower molecular weight, consistent with their inability to form tetramers, while TAp53γ and Δ40p53γ exhibited additional aggregation behaviour. Of the Δ133 and Δ160 isoforms, the α-C-terminus showed only a slight aggregation tendency, while both β- and γ-C-termini displayed very strong aggregation, as did the TAp53Ψ and TAΔp53 isoforms.

We further investigated the behaviour of these isoforms (aggregation is independent of Myc-tag position; **Figure 4—figure supplement 1**) by native PAGE with similar results, except that in this experimental setting also the Δ133p53α and Δ160p53α isoforms showed strong aggregation (**Figure 4E**). Finally, we determined the relative amounts of protein in the soluble and insoluble fraction after expression in H1299 cells. In this assay, again all Δ133 and Δ160 isoforms were found to a large degree in the insoluble fraction (**Figure 4F**, **Figure 4—figure supplement 2**). Δ40p53γ was also mainly found in the pellet while TAp53γ was mainly soluble as were the TAp53φ and TAΔp53 isoforms. As the γ-C-terminus seems to promote aggregation, we measured the tendency of the dye Thioflavin T (ThT) (**Cino et al., 2016**) to bind to a peptide derived from the γ-C-terminus and compared it to a peptide derived from the β-C-terminus. Thioflavin T binding was confirmed for the γ-peptide, while the β-peptide did not show binding above the background level, suggesting that indeed the γ-C-terminus has a high aggregation propensity in vitro and in cellulo (**Figure 4G**).

Aggregation is a function of the sequence but also of the concentration. In a cellular system, chaperones are present that keep proteins with open hydrophobic stretches soluble, prevent them from aggregation and, if this is impossible, associated E3 ligases will degrade the misfolded protein. To test the dependence on the concentration, we expressed several isoforms under a minimal CMV promoter (selected artificial promoter, p5RPU.Myc [**Brown et al., 2017**], was cloned into pcDNA3.1 plasmid) which results in reduced expression levels in H1299 cells. Using this system, also Δ133p53α,β could indeed be kept soluble (**Figure 4—figure supplement 3**), suggesting that aggregation will only become a problem if the expression level surpasses a certain threshold level. To further test this hypothesis, we overexpressed Δ40p53α, Δ133p53α, and Δ133p53β under the strong promoter in H1299 cells and treated the cells with different HSP70 and HSP90 inhibitors. No effect was seen for Δ40p53α which remained soluble. For both Δ133p53α and Δ133p53β, however, the percentage of the insoluble fraction increased relative to the treatment with only DMSO, suggesting that inhibition of chaperones results in increased aggregation (**Figure 4—figure supplement 3**).

## Interaction with chaperones

The experiments described above suggested that chaperones are involved in keeping aggregation-prone p53 isoforms soluble. The characterization so far predicted that several of the p53 isoforms have exposed hydrophobic patches due to misfolding of the DBD or as part of isoform-specific new APRs. We tested the interaction of p53 isoforms with chaperones by transiently transfecting H1299 cells with the individual p53 isoforms and then immunoprecipitating endogenous HSC/HSP70. Western blot analysis with an α-Myc antibody was used to detect co-immunoprecipitation (co-IP) of the N-terminally myc-tagged p53 isoforms. A strong signal was detected for the cancer mutant p53V157F (**Figure 5A**, **Figure 5—figure supplement 1**). Further interactions were seen for Δ40p53γ, Δ133α,β,γ, TAp53φ, and TAΔp53. Only weak signals occurred for wtp53, Δ40p53α, Δ40p53β, and p53R273H in agreement with the interpretation that these isoforms and mutants do not have accessible hydrophobic patches or APRs.

We wanted to further investigate if expression of p53 isoforms creates cellular stress and triggers the expression of chaperones. To study this question, we measured the transcriptional activity on

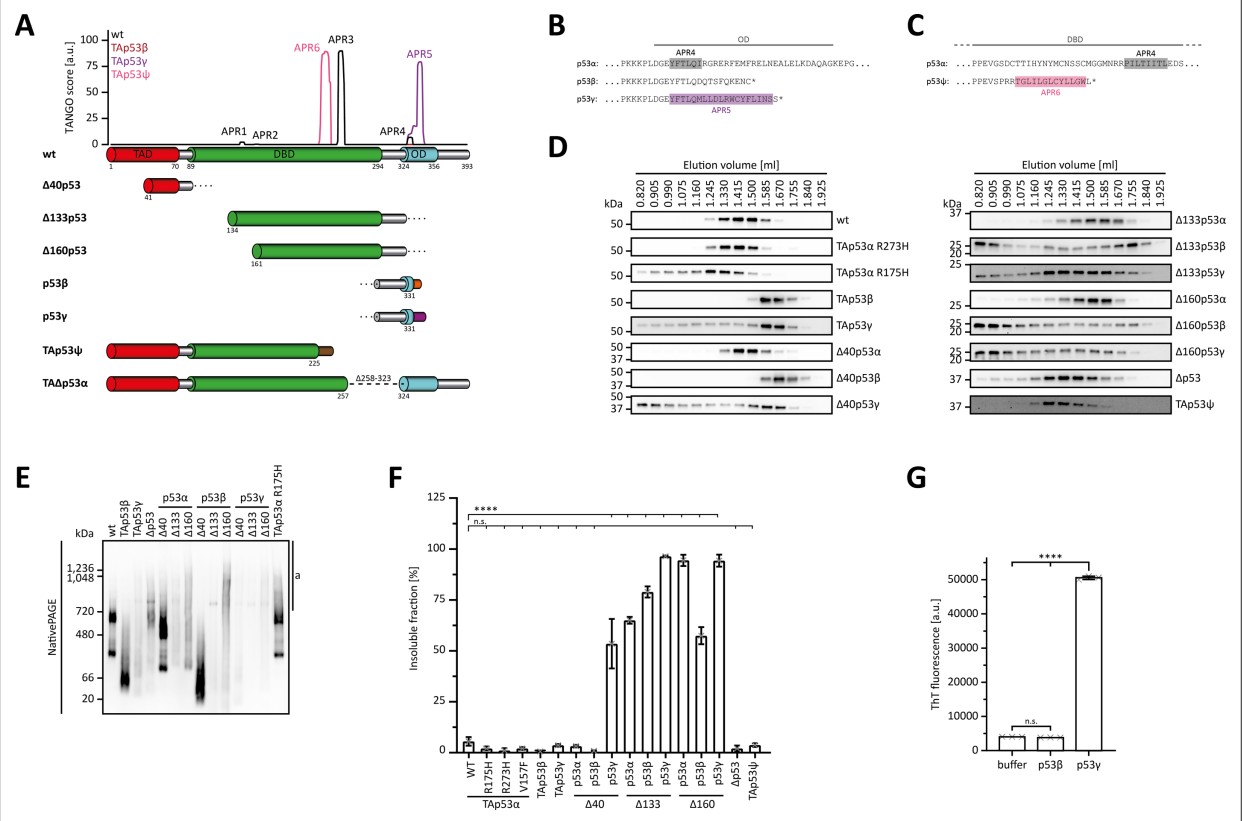

**Figure 4.** Aggregation and solubility. (**A**) Predicted aggregation propensity of TAp53α (black) and the C-terminal isoforms TAp53β (auburn), TAp53γ (purple), and TAp53$\phi$ (pink). TAp53α is predicted to have a total of four aggregation-prone regions (APR1–4) sharing the first three with TAp53β and TAp53γ. TAp53γ and TAp53$\phi$ each introduced a novel APR (APR5 and APR6) with their specific C-termini. A schematic illustration of the boundaries of all N- and C-terminal p53 isoforms is shown below the graph. Position and length of the novel sequences introduced by the C-terminal are shown with the corresponding colours. All in silico predictions were run using the TANGO algorithm with the following parameters: pH 7.5, 150 mM ionic strength and at a temperature of 37°C. (**B**) Sequence alignment of p53α with the p53β- and p53γ-specific C-termini. The boundaries of the OD are indicated by the grey line and the predicted APRs from (**A**) are highlighted in the respective colours. (**C**) Sequence alignment of p53α and the p53$\phi$-specific C-terminus. The boundaries of the DNA-binding domain (DBD) are indicated by the grey line and the predicted APRs from (**A**) are highlighted in the respective colours. (**D**) Analytical SEC of p53 isoforms and cancer-related mutants. H1299 cells were transiently transfected with the N-terminally Myc-tagged proteins and cell lysates were loaded onto a Superose 6 SEC column. Both lysis and running buffer contained 20 mM CHAPS. Collected fractions were analysed for p53 by WB using an α-Myc antibody. An elution volume of 0.820 ml corresponds to the void volume of the column (2.4 ml bed volume). (**E**) BN–PAGE of p53 isoforms and the cancer-related R175H mutant. H1299 cells were transiently transfected with the N-terminally Myc-tagged proteins. Cell lysates were subsequently analysed by BN–PAGE and SDS–PAGE (shown in *Figure 4—figure supplement 1*) followed by WB using α-Myc antibody for detection. High molecular weight species corresponding to aggregates are marked by 'a'. (**F**) Solubility assay of p53 isoforms and cancer-related mutants. H1299 cells were transiently transfected with the indicated Myc-tagged proteins and lysed in a buffer supplemented with Triton X-100. Soluble and insoluble components were separated by centrifugation. The insoluble fraction in the pellet was solubilized with a buffer supplemented with SDS. Samples of both fractions were analysed by WB using αMyc antibody. (**G**) Thioflavin T (ThT) fluorescence assay of the p53β (aa 322–341) and p53γ (322–346) C-termini. The peptides, solubilized in denaturation buffer, and denaturation buffer only as a control, were diluted 20-fold in assay buffer supplemented with ThT and incubated at 37°C for 45 min. The final concentration of peptides and ThT was 20 and 25 µM, respectively. (**F, G**) The bar diagram shows the mean ThT fluorescence and error bars the corresponding SD ($n$ = 3). Statistical significance was assessed by ordinary one-way ANOVA (n.s.: $p > 0.05$, ****$p \leq 0.0001$).

The online version of this article includes the following source data and figure supplement(s) for figure 4:

**Source data 1.** Uncropped Western blots.

**Figure supplement 1.** Investigation of the effect of the Myc-tag.

**Figure supplement 1—source data 1.** Uncropped Western blots.

**Figure supplement 2.** Solubility of p53 isoforms.

**Figure supplement 2—source data 1.** Uncropped Western blots.

**Figure supplement 3.** Chaperone interaction of p53 isoforms.

**Figure supplement 3—source data 1.** Uncropped Western blots.

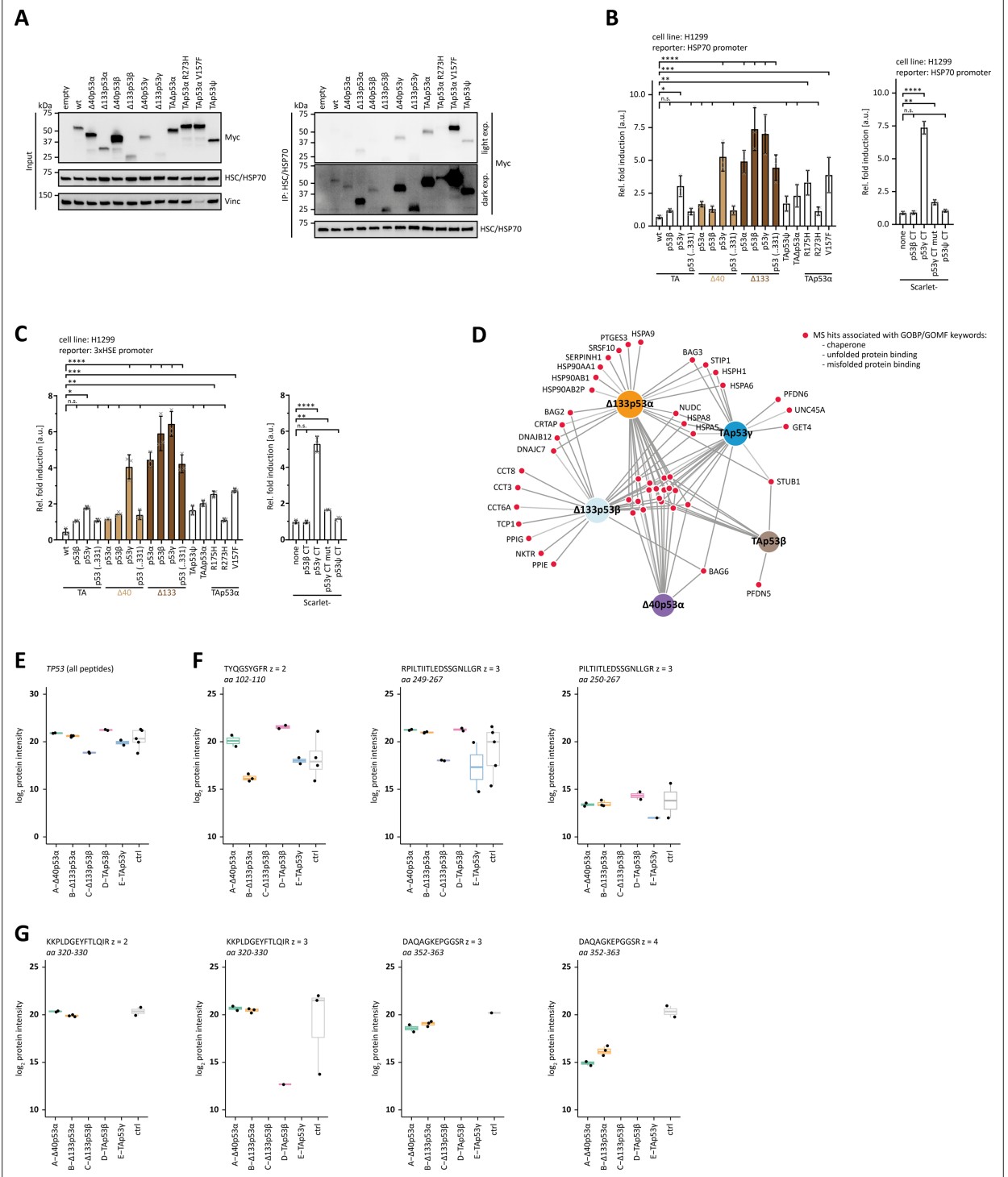

**Figure 5.** Chaperones. (**A**) Co-immunoprecipitation of p53 isoforms and cancer-related mutants with endogenous HSC/HSP70. H1299 cells were transiently transfected with either empty vector or the indicated N-terminally Myc-tagged p53 variants. HSC/HSP70 was immunoprecipitated (IP) with an αHSC/HSP70 antibody. Input and IP samples were subsequently analysed by WB using αHSC/HSP70 and α-Myc antibody to detect HSC/HSP70 and p53 variants, respectively. A light and dark exposure of the IP samples detected with α-Myc antibody is shown. Vinculin served as a loading control for the input samples. (**B**) Luciferase reporter assay of p53 isoforms and mutants as well as the indicated Scarlet fusion proteins on the HSP70 promoter. H1299 cells were transiently transfected with the respective luciferase reporter plasmids and the N-terminally Myc-tagged proteins. (**B**) p53 (331) contains only the C-terminal part common to p53α, p53β, and p53γ. The fluorescent protein Scarlet was fused with the C-termini of p53 isoforms. Scarlet alone served as a negative control. (**C**) Luciferase reporter assay of p53 isoforms and mutants as well as the indicated Scarlet fusion proteins on

*Figure 5 continued on next page*

*Figure 5 continued*

the heat-shock element (HSE) promoter (containing three repeats of heat shock element). H1299 cells were transiently transfected with the respective luciferase reporter plasmids and the N-terminally Myc-tagged proteins. (**C**) p53 (331) contains only the C-terminal part common to p53α, p53β, and p53γ. The fluorescent protein Scarlet was fused with the C-termini of p53 isoforms. Scarlet alone served as a negative control. (**B, C**) The bar diagram shows the mean fold induction relative to the empty vector control and error bars the corresponding SD ($n = 3$). Statistical significance was assessed by ordinary one-way ANOVA (n.s.: $p > 0.05$, *$p ≤ 0.05$, **$p ≤ 0.01$, ***$p ≤ 0.001$, ****$p ≤ 0.0001$). (**D**) Chaperones and other proteins associated with binding un-/misfolded proteins are shown, which are significantly enriched in the mass spectrometry analysis of p53 isoforms. The red nods represent the identified proteins assigned by the grey lines to the p53 isoforms they were enriched for. Significant hits were determined by setting the parameters: $\log_2$ enrichment greater than or equal to 0.5 and p-value less than 0.05 and proteins were filtered for the keywords ('chaperone', 'unfolded protein binding', and 'misfolded protein binding') in the GOBP (Gene Ontology Biological Process) and GOMF (Gene Ontology Molecular Function) terms. The plot was generated using DiVenn (v2.0). (**E**) Results from the second mass spectrometry experiment, focused on the quantification of p53 peptide precursors. As the biotin ligase was directly fused to p53, p53 peptides are quantified for all samples ($\log_2$-transformed protein LFQ intensity). Negative controls represent data from unbiotinylated lysates (no biotin added). (**F**) Peptides from the DNA-binding domains spanning residues 249–267 were quantified for all samples ($\log_2$-scaled precursor intensity), consistent with this peptide being part of all investigated isoforms. A peptide spanning amino acids 102–110 was absent or beyond the detection limit in the Δ133p53β sample but quantified in all other samples. (**G**) Peptides originating from the oligomerization domain were reliably quantified in samples of isoforms containing a full-length oligomerization domain ($\log_2$-scaled precursor intensity), but not in samples of isoforms containing the β- and γ-C-termini (except for one low intense TAp53β precursor ion).

The online version of this article includes the following source data and figure supplement(s) for figure 5:

**Source data 1.** Uncropped Western blots.

**Figure supplement 1.** Chaperones.

**Figure supplement 1—source data 1.** Uncropped Western blots.

**Figure supplement 2.** Mass spectrometry based proteomics.

a reporter plasmid expressing luciferase under an HSP70 promoter (*Figure 5B*, *Figure 5—figure supplement 1*) and expressed at the same time different p53 isoforms. If the expression of these isoforms induced cellular stress which would result in the upregulation of chaperone expression, an increase in the luciferase activity should be detectable. These experiments showed that expression of p53 isoforms containing a γ-C-terminus, as well as all Δ133p53 versions, led to a strong activation of the HSP70 promoter. Elevated levels were also detected for TAΔp53, p53R175H, and p53V157F, but not for the DNA-binding interface mutant p53R273H. In addition, we investigated the effect of the isoform-specific β- and γ-C-termini by fusing them to the protein Scarlet (*Gadella et al., 2023*). We compared the activity on the HSP70 promoter in cells expressing these fusion proteins with the activity in cells expressing Scarlet itself. The results showed that the γ-C-terminus strongly induced activity, but not the β-C-terminus. Responsible for the aggregation-promoting effect of the γ-C-terminus (which according to the Waltz predictor is potentially amyloidogenic [*Beerten et al., 2015*]) is presumably its hydrophobic nature. To investigate this hypothesis, we created the Y327A, L330A, L334A, and L336A mutations in the γ-C-terminus which indeed reduced the luciferase expression to background levels.

Overall, these results suggest that luciferase expression via the HSP70 promoter was induced by the overexpression of certain p53 isoforms, suggesting that their only partially folded state and/or open hydrophobic patches induce a cellular stress response. To exclude the possibility that the p53 isoforms induce luciferase expression by direct DNA binding (although our experiments described above already showed that only wtp53 and Δ40p53α can bind DNA strongly), we repeated the same experiments with a reporter plasmid having three copies of a heat-shock element (HSE) in front of a SV40 promoter and which therefore – in contrast to the full HSP70 promoter (*Agoff et al., 1993*) – does not contain any p53 REs. These experiments confirmed the data obtained with the HSP70 promoter, strongly suggesting that it is the isoform induced cellular stress that triggers the expression of chaperones (*Figure 5C*).

We further investigated the interaction of selected p53 isoforms with chaperones by mass spectrometry-based proteomics. For this purpose, we created stable U2OS cell lines in which the isoforms were fused to the C-terminus of the TURBO-ID biotin ligase with their expression being under control of a Tet Response Element. After induction of expression of the isoforms (with tetracycline) and initiation of biotinylation, we isolated biotinylated proteins with streptavidin-coated magnetic beads and used a Q Exactive HF coupled to an easy nLC 1200 mass spectrometer (Thermo Fisher Scientific) to analyse proteins that were in close proximity to these selected isoforms. The results were

filtered for the terms 'chaperone', 'unfolded binding', and 'misfolded binding'. *Figure 5D* shows that all isoforms interact with a common set of proteins (*Supplementary file 1*). In addition to this common set, TAp53β and Δ40p53α showed only very few additional interaction partners. This situation was very different for Δ133α, Δ133β, and TAp53ɣ, which all showed close proximity to many more proteins involved in binding to un- or misfolded proteins. These data further confirm that Δ133p53 isoforms and isoforms with a ɣ-C-terminus contain hydrophobic patches and that their expression triggers cellular stress.

p53 itself was one of the most prominent proteins found in the interactome of all five investigated isoforms (*Supplementary file 1*). This is expected as the TurboID ligase was fused to the N-terminus of all isoforms, which results in efficient self-biotinylation. To investigate if these isoforms also interact with wild-type p53, we focused a new mass spectrometry experiment on the identification of p53 peptides and analysed the interactome data for peptide precursors that can be quantified only in wild-type p53 but not in the specific isoform (*Figure 5E–G*, *Figure 5—figure supplement 2*). For Δ133p53α, we could quantify a peptide precursor between amino acids 102 and 110 that is highly likely to originate from wild-type p53. This result is not surprising as Δ133p53α contains a full oligomerization domain capable of forming mixed tetramers with wild-type p53. These peptides were absent or beyond the detection limit in the Δ133p53β sample, in line with the interpretation that this isoform cannot hetero-oligomerize with wild-type p53. Peptides spanning the region amino acids 319–333 were only quantified in samples of Δ40p53α and Δ133p53α which contain the full-length oligomerization domain themselves, which makes it impossible to determine if these peptides originate from wild-type p53 or from the investigated p53 isoform. For Δ133p53β, TAp53β, and TAp53ɣ, we did not find any peptides in this region (with the exception of a precursor of very low intensity in a single Δ133p53β sample), supporting the interpretation that these isoforms do not interact with wild-type p53. For Δ40p53α, no wild-type peptides originating from the first 40 amino acids were detected despite the presence of a full oligomerization domain. Overall, these data show that Δ133p53α can form mixed tetramers and can thus act dominantly negative towards wild-type p53.

## Some p53 isoforms can inhibit wt-p53, p63, and p73 by different mechanisms

The tetrameric state of p53 is the basis for the dominant negative effect of cancer mutations that inactivate DNA binding. The incorporation of one or more mutated p53 monomers into the tetramer weakens the binding to DNA. We wanted to investigate how the co-expression of selected p53 isoforms affects the activity of wtp53, p73, and p63. Co-expression of known cancer mutants strongly inhibited the activity of wtp53 as expected, as did co-expression of all α-isoforms (*Figure 6A*, *Figure 6—figure supplement 1*). Expression of the Δ40p53α isoform can inhibit the activity of wtp53 also by blocking DNA binding (promoter squelching). Without a full TAD, this isoform is transcriptionally strongly suppressed. Co-expression of any isoform that is incapable of forming tetramers did not show any inhibitory effect. This includes all β- and ɣ-C-termini regardless of the N-terminal version as well as p53 φ and TAΔp53α. The inability of TAΔp53α to inhibit wtp53 seems surprising as it contains a truncated DBD but a full-length OD, in principle allowing it to form mixed tetramers with wtp53 (*García-Alai et al., 2008*). However, the deleted sequence includes the NLS of the protein, which traps this isoform in the cytoplasm and prevents a dominant negative effect as also previously described (*Chan and Poon, 2007*).

Previous studies had shown that p73 and p63 do not interact with p53 via their ODs (*Coutandin et al., 2009*; *Joerger et al., 2009*) but form mixed p63/p73 tetramers with a tetramer built from a p63 dimer and a p73 dimer being the thermodynamically most stable species (*Gebel et al., 2016*). Co-expression of TAp73α with ΔNp63α, a p63 isoform that lacks the transactivation domain and therefore shows very little transcriptional activity on most promoters, inhibits p73's activity, most likely by promoter squelching as previously described (*Gebel et al., 2016*; *Figure 6B*, *Figure 6—figure supplement 1*). Co-expression with ΔNp63αR304W, a mutant form incapable of binding DNA, however, showed that inhibition can also be due to the formation of mixed tetramers with decreased DNA-binding affinity. Co-expression with Δ40p53α again strongly inhibited the activity of TAp73α, not through formation of mixed tetramers but by promoter squelching which could be confirmed from the inability of a R273H mutant of Δ40p53α to inhibit p73. All other p53 isoforms tested had no effect, consistent with their inability to bind to DNA and replace p73. The only exceptions were the Δ133

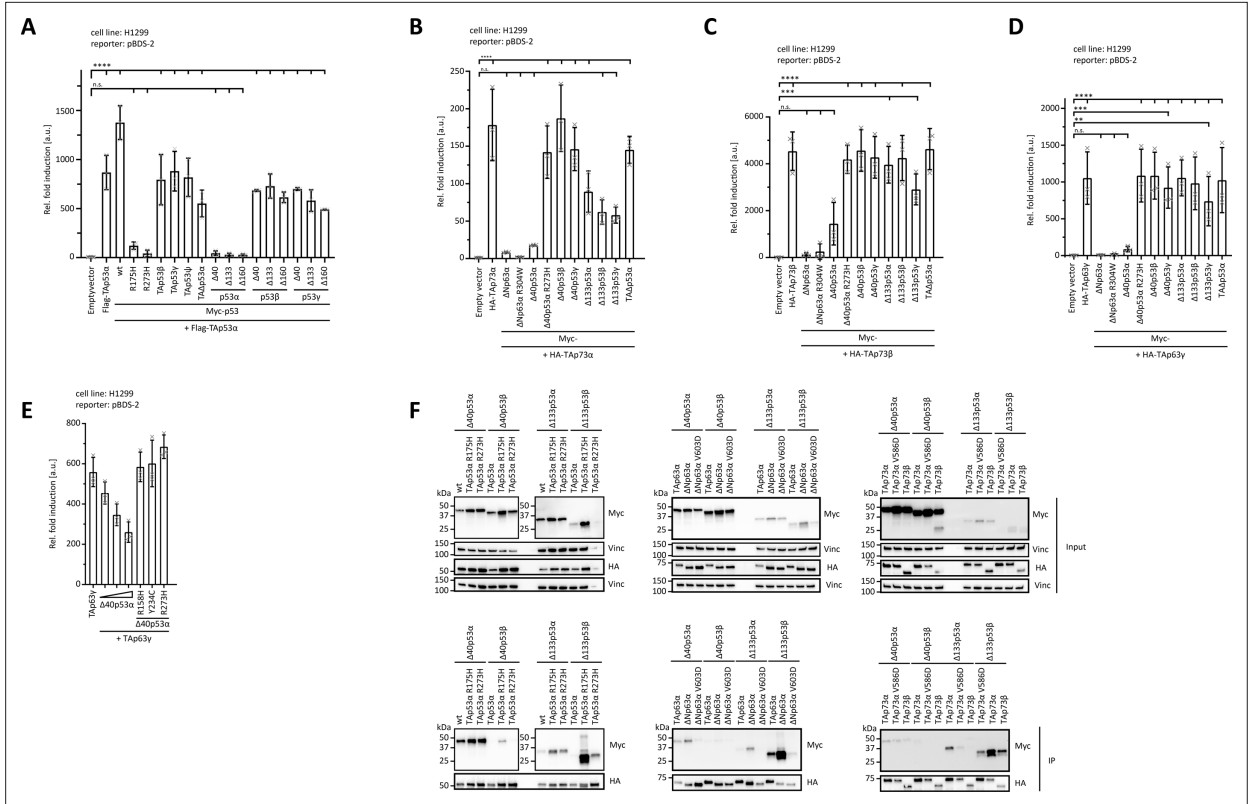

**Figure 6.** Inactivation of p53 family members. (**A**) Luciferase reporter assay of TAp53α in combination with p53 isoforms and cancer-related mutants on the pBDS-2 reporter. H1299 cells were transiently transfected with the respective luciferase reporter plasmids, the N-terminally Flag-tagged TAp53α alone or together with the N-terminally Myc-tagged p53 variants. Luciferase reporter assay of TAp73α (**B**), TAp73β (**C**), or TAp63γ (**D**) in combination with p53 and p63 isoforms and cancer-related mutants on the pBDS-2 reporter. H1299 cells were transiently transfected with the respective luciferase reporter plasmids, the N-terminally HA-tagged TAp73α (**B**), TAp73β (**C**), or TAp63γ (**D**) alone, empty vector or together with the N-terminally Myc-tagged p53/p63 variants. (**A–D**) The bar diagram shows the mean fold induction relative to the empty vector control and error bars the corresponding SD (*n* = 3). Statistical significance was assessed by ordinary one-way ANOVA (n.s.: p > 0.05, *p ≤ 0.05, **p ≤ 0.01, ***p ≤ 0.001, ****p ≤ 0.0001). (**E**) Luciferase reporter assay of TAp63γ in combination with the Δ40p53α isoforms and cancer-related mutants on the pBDS-2 reporter. H1299 cells were transiently transfected with the respective luciferase reporter plasmids, the N-terminally Myc-tagged TAp63γ alone or together with the N-terminally Myc-tagged Δ40p53α variants. (**F**) Conformation-specific immunoprecipitation (Conf-IP) of p53, p63, or p73 isoforms and p53 variant (Δ40p53α, Δ40p53β, Δ133p53α, and Δ133p53β). H1299 cells were transiently transfected with N-terminally HA-tagged p53, p63, or p73 isoforms and N-terminally Myc-tagged p53 variants (Δ40p53α, Δ40p53β, Δ133p53α, and Δ133p53β). p53, p63, or p73 isoforms were immunoprecipitated (IP) with α-HA. p53α isoforms mostly hetero-tetramerize with different p53α isoforms, but co-aggregate with remaining p53 isoforms. p63 and p73 isoforms interact with p53 isoforms by co-aggregation. Input and IP samples were subsequently analysed by WB using αMyc antibody.

The online version of this article includes the following source data and figure supplement(s) for figure 6:

**Source data 1.** Uncropped Western blots.

**Figure supplement 1.** Inactivation of p53 family members.

**Figure supplement 1—source data 1.** Uncropped Western blots.

isoforms that showed an intermediate degree of inhibition. In a previous study, we had shown that the α-C-termini of p73α and p63α can confer co-aggregation with destabilized p53 cancer mutants such as p53R175H (*Kehrloesser et al., 2016*). To probe if the observed inhibition by the Δ133 isoforms could be due to co-aggregation with this α-C-terminus, we repeated the experiments with TAp73β which lacks this aggregation-prone sequence. The results mirror the data obtained with TAp73α, with the exception that the inhibitory effect of the Δ133p53 isoforms was indeed strongly reduced (*Figure 6C*, *Figure 6—figure supplement 1*).

We also investigated the effect of p53 isoforms on p63's activity. As TAp63α forms inactive dimers (*Deutsch et al., 2011*), we used TAp63γ, a constitutive tetrameric and active isoform (*Yang et al., 1998*). The results are very similar to the data obtained with TAp73β, showing an inhibitory effect through promoter squelching only for Δ40p53α (*Figure 6D*, *Figure 6—figure supplement 1*). We

further confirmed the interpretation that promoter squelching is the relevant mechanism by measuring the transcriptional activity of TAp63γ in the presence of increasing amounts of Δ40p53α which resulted in a concentration-dependent inhibition. The presence of Δ40p53α harbouring mutations in the DBD that inhibit DNA binding did not affect the transcriptional activity of TAp63γ (*Figure 6E*).

Finally, we used co-IP to investigate direct physical interactions between selected p53 isoforms and wtp53, p63 and p73. We transfected H1299 cells transiently with N-terminally HA-tagged p53, p63, or p73 isoforms and with N-terminally Myc-tagged p53 variants (Δ40p53α, Δ40p53β, Δ133p53α, and Δ133p53β). p53, p63, or p73 isoforms were IP with an α-HA antibody and co-IP of p53 variants was probed with Western blots using an α-Myc antibody (*Figure 6F*). Δ40p53α interacted strongly with wtp53 as well as with the p53R175H and p53R273H cancer mutants, while Δ40p53β interacted weakly only with p53R175H, suggesting that these interactions are mediated by the OD. Similarly, interaction with all three p53 variants resulted in co-IPs with Δ133p53α. Interaction with wtp53 was weakest, showing that Δ133p53α can drag wtp53 into insoluble aggregates, but not very efficiently. Of the Δ133p53β co-IPs, the strongest was with the p53R175H cancer mutant which forms co-aggregates. For p63, we tested the dimeric TAp63α, the open tetrameric ΔNp63α and the ΔNp63αV603D version in which the aggregation-prone C-terminus is mutated to reduce the aggregation propensity (*Kehrloesser et al., 2016*). None of them showed interaction with Δ40p53α or Δ40p53β. The same was true for the co-IP with Δ133p53α. A co-IP could, however, be detected for ΔNp63α and Δ133p53β, which could be reduced by the V603D mutation in the p63 C-terminus, showing that this interaction is based on co-aggregation. Virtually the same results were obtained with co-IP experiments with different p73 isoforms. TAp73α, which is a constitutively open and tetrameric isoform (*Luh et al., 2013*), shows co-aggregation with Δ133p53α and Δ133p53β, which can get reduced by a V586D mutation in the α-C-terminus that reduces the aggregation propensity (*Kehrloesser et al., 2016*). Likewise, TAp73β, which lacks the α-C-terminus, shows a strongly reduced co-IP.

## Discussion

The p53 protein family plays very important roles in genetic quality control, developmental programmes and in tumour suppression (*Levine, 2020*). The discovery of p63 and p73 has shown that both proteins can exist in many different isoforms. In the case of p63, very clear biological functions have been assigned to two of these, namely ΔNp63α which plays an important role in organizing the chromatin landscape in the basal cells of epithelial tissues (*Yang et al., 1998*; *Kouwenhoven et al., 2015*) and TAp63α which is a crucial factor for maintaining the genetic quality in oocytes (*Deutsch et al., 2011*; *Suh et al., 2006*). For p73 functions in the development and maintenance of neurons, the development of ciliated cells (*Marshall et al., 2016*) and in tumour suppression (*Logotheti et al., 2022*) has been described involving both TA- and ΔN-isoforms with the α- and the β-C-termini. Characteristic for these isoforms is that they contain fully folded domains and differ only in the complete absence or presence of such domains. The p53 isoforms show a different picture. With the exception of Δ40p53α, which lacks parts of the unstructured transactivation domain, all are characterized by containing only parts of the DBD and/or the OD. As protein folding of single domains is in general a cooperative process, this predicts that these domains with missing secondary structure elements are largely unfolded and not capable of fulfilling a normal cellular function. Here, we have used multiple experiments to provide a comprehensive characterization of these p53 isoforms. Our data demonstrate that high-affinity DNA binding requires both a folded DBD as well as a folded OD. Only Δ40p53α fulfils these requirements and might have its own meaningful biological function, potentially as a transcriptional inhibitor of wtp53, similar to ΔNp63 and ΔNp73 isoforms which can inhibit wtp53 by promoter squelching. The potential role of Δ40p53α could be that of fine tuning of wtp53's activity as the formation of hetero-complexes with a reduced number of TADs influences not only the transcriptional activity directly but potentially also indirectly as a reduced number of TADs can reduce the Mdm2-dependent degradation rate. This can lead to an even increased overall transcriptional activity as shown for titrations of TAp63 and ΔNp63 isoforms (*Yang et al., 1998*). All other p53 isoforms do not bind strongly to DNA either due to lack of a functional DBD, a functional OD or even both. Instead, the unfolded/partial folded states, in particular of the DBD, as well as the hydrophobic nature of the γ-C-terminus, increase cellular stress as evidenced by the upregulation of chaperones and the association of chaperones and proteins involved in interaction with unfolded/misfolded proteins with these isoforms. Below a certain expression level, the cellular chaperone systems seem to be able to keep these isoforms soluble, but

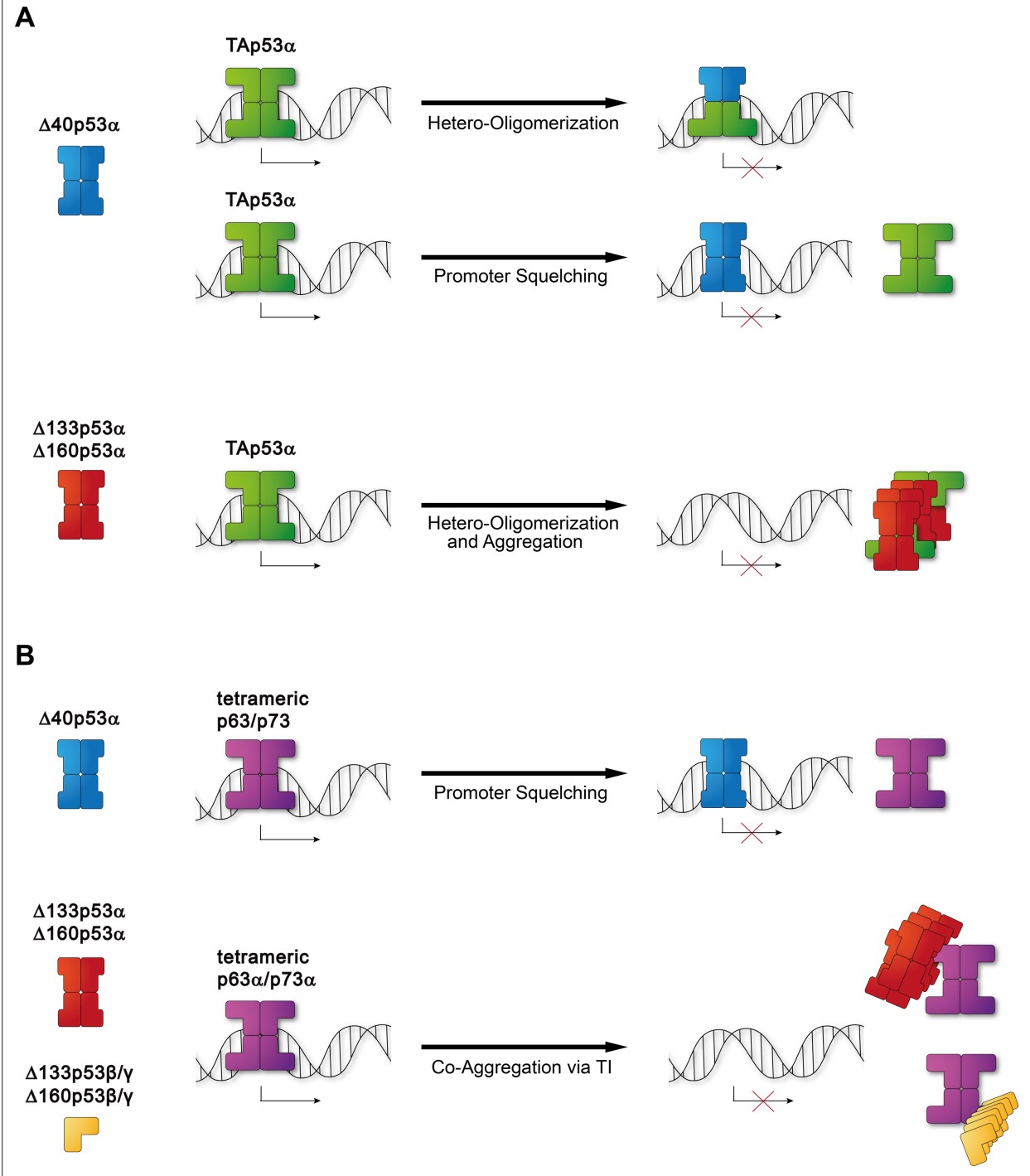

**Figure 7.** Mechanisms of inactivation within the p53 family. p53 isoforms have the potential to inactivate p53 family members via different mechanisms. (**A**) Either p53α isoforms form inactive hetero-tetramers (and/or aggregate) with wtp53 (Δ40p53α, Δ133p53α, and Δ160p53α) and therefore inhibit wtp53 or inactive wtp53 by promoter squelching (Δ40p53α). (**B**) Furthermore, Δ40p53α isoform can as well inactivate tetrameric p63/p73 via promoter squelching. DNA-binding domain (DBD)-truncated aggregating p53 isoforms can co-aggregate with tetrameric p63α and p73α isoforms, but not with the closed TAp63α dimer.

above this level they accumulate in insoluble aggregates. While the lack of crucial functional domains prevents well-defined physiological roles of these isoforms (with the potential exception of Δ40p53α as mentioned above), they have the potential to inactivate members of the p53 protein family by forming inactive hetero-complexes with wtp53 (Δ133p53α and Δ160p53α), by direct co-aggregation

with α-isoforms of p63 and p73 (Δ133p53α,β,ɣ and Δ160p53α,β,ɣ) or promoter squelching (Δ40p53α). The exception to this rule is TAΔp53α, which aggregates (*García-Alai et al., 2008*) and could in theory bind to wtp53 via its OD, but lacks an NLS and therefore cannot be imported into the nucleus to execute its dominant negative effect. Forced translocation of TAΔp53α into the nucleus was shown to establish a dominant negative effect, demonstrating that the lack of the NLS is the main impairment of a negative effect on wtp53 (*Chan and Poon, 2007*). The potential interaction mechanisms of p53 isoforms with members of the p53 family are summarized in *Figure 7*.

Through these mechanisms, expression of many of the p53 isoforms can have detrimental effects, and the literature on these isoforms so far shows a clear correlation with cancer. The least effect is to be expected from Δ40p53β, which is only little more than an isolated p53 DBD and Δ40p53ɣ. This isoform, however, has the ability to aggregate and co-aggregate with other proteins due to the very hydrophobic nature of the ɣ-C-terminus. For all other isoforms, severe effects – increasing with the cellular concentration – can be expected. Depending on which cell types this expression occurs in, different overall physiological effects (cancer, neurodegeneration) can result. While this study is focused on the effects that truncations have on the folded domains of p53, it should be mentioned that deletion of the unfolded N-terminal as well as C-terminal domains also impairs the normal function of p53 (*Dyson and Wright, 2025*). These domains are the target of a large number of post-translational modifications and are important for the interaction with regulatory proteins, for example p300 and Mdm2 in the transactivation domain.

The aggregation potential of many of the p53 isoforms has been noticed in several other publications as well. The lab of Curtis Harris has shown by mass spectrometry analysis following co-IP of FLAG-tagged Δ133p53α in MRC-5 fibroblasts that Δ133p53α interacts with members of the Hsp70 chaperone families as well as the chaperone-associated E3 ubiquitin ligase STUB1 as well as BAG2 and BAG5 (*Horikawa et al., 2014*), similar to our Turbo-ID-based proteomics study reported here (*Figure 5D*). Moreover, Vieler et al. stated that they observed increased aggregation of Δ133p53 without presenting any data (*Vieler and Sanyal, 2018*), whereas a computational study of the Δ133p53 DBD suggested a lower stability correlated with a high tendency for aggregation (*Lei et al., 2019*). Experimentally, the aggregation behaviour of Δ133p53β has been investigated by *Arsic et al., 2021*. They also found that this isoform interacts with the CCT chaperone complex and that depletion of this complex further promotes aggregation. Direct experimental evidence for the instability and aggregation tendency of the TAΔp53 isoform was provided by the Fersht laboratory (*García-Alai et al., 2008*). They confirmed that this isoform cannot bind to the p21 promoter and in mixed states significantly reduces the DNA-binding affinity of wtp53.

All the data suggesting that deletion of parts of the DBD results in the inability to bind DNA, the exposure of aggregation-prone peptide sequences and aggregation as well as co-aggregation with other cellular factors are supported by a large literature on the effect of structural mutations in the p53 DBD. Numerous studies have shown that the only metastable DBD of p53 gets further destabilized by these mutations leading to loss of DNA binding and aggregation behaviour (*Xu et al., 2011*; *Wang and Fersht, 2015*). If single point mutations in the DBD can have that strong effects, it is more than likely that deletions of entire parts of the domain will have even stronger effects – consistent with all the data reported here.

In conclusion, the effects seen in cell culture as well as in patients with overexpression of different p53 isoforms should be interpreted on the background of the aggregation propensity as well as a potential overload of the cellular chaperone system instead of any specific interaction of these isoforms with proteins or DNA.

# Materials and methods

**Key resources table**

| Reagent type (species) or resource | Designation | Source or reference | Identifiers | Additional information |
|---|---|---|---|---|
| Cell line (human) | H1299 | ATCC | CRL-5803 | |
| Cell line (human) | SAOS | ATCC | HTB-85 | |

*Continued on next page*

*Continued*

| Reagent type (species) or resource | Designation | Source or reference | Identifiers | Additional information |
|---|---|---|---|---|
| Cell line (human) | U-2OS Flp-In T-Rex | Gift from C. Behrends | | Generated according to the protocol of Thermo Fisher Scientific using the Flp-in T-Rex core lot (cat number K650001) |
| Antibody | anti-Myc | Clone 4A6, Merck KGaA | 05-724 | 1:1000 |
| Antibody | anti-HA goat polyclonal | Bethyl Laboratories | A190-107A | 1:1000 |
| Antibody | anti-p53 (DO-1) | Santa Cruz Biotechnologies | sc-126 | 1:1000 |
| Antibody | anti-p53 (pAb240) | Santa Cruz Biotechnologies | sc-99 | 1:1000 |
| Antibody | anti-Vinculin | Santa Cruz Biotechnology Clone 7F9 | sc-73614 | 1:1000 |
| Antibody | anti-GST-HRP | GE Healthcare Life Sciences | RPN1236 | 1:10,000 |
| Antibody | anti-Streptavidin-HRP | Sigma-Aldrich Chemie GmbH | S2438 | 1:10,000 |
| Antibody | anti-Flag | Sigma-Aldrich Chemie GmbH Clone M2 | F3165 | 1:1000 |
| Antibody | anti-HSP70/HSC70 | Santa Cruz Biotechnology Clone W27 | sc-24 | 1:1000 |
| Antibody | Peroxidase AffiniPure Goat Anti-Mouse IgG (H+L) | Jackson ImmunoResearch | 115-035-062 | 1:2000 |
| Antibody | Peroxidase AffiniPure Donkey Anti-Goat IgG (H+L) | Jackson ImmunoResearch | 705-035-003 | 1:2000 |
| Recombinant DNA reagent | pcDNA3.Myc | Thermo Fisher Scientific | | Derivative of pcDNA3.1(+) (cat number V79020) |
| Recombinant DNA reagent | p5RPU.Myc | Thermo Fisher Scientific | | Modified from pcDNA3.Myc |
| Recombinant DNA reagent | pcDNA3.HA | Thermo Fisher Scientific | | Derivative of pcDNA3.1(+) (cat number V79020) |
| Recombinant DNA reagent | pcDNA3.Flag | Thermo Fisher Scientific | | Derivative of pcDNA3.1(+) (cat number V79020) |
| Recombinant DNA reagent | pGL3-Basic vector | Promega | | |
| Recombinant DNA reagent | pRL-CMV | Promega | E2261 | |
| Recombinant DNA reagent | pBV-Luc BDS-2 3x WT | Addgene | Plasmid #16515 | |
| Recombinant DNA reagent | pGL2 basic vector with Mdm2 promoter | Promega | | Derivative of pGL2 (GenBank X65323) |
| Recombinant DNA reagent | pGL3 basic vector with p21 promoter | Promega | | Derivative of pGL3 (GenBank U47295.2) |
| Recombinant DNA reagent | pGL3 basic vector with HSP70 promoter | Promega | | Derivative of pGL3 (GenBank U47295.2) |
| Recombinant DNA reagent | pGL3 basic vector with HSP70/3x heat-shock elements (HSEs) | Promega | | Derivative of pGL3 (GenBank U47295.2) |
| Recombinant DNA reagent | pcDNA5/FRT/TO | Thermo Fisher Scientific | V652020 | |
| Recombinant DNA reagent | pOG44 | Thermo Fisher Scientific | V600520 | |

*Continued on next page*

*Continued*

| Reagent type (species) or resource | Designation | Source or reference | Identifiers | Additional information |
|---|---|---|---|---|
| Recombinant DNA reagent | pGEX-6P-2 | Cytiva | 28954650 | |
| Recombinant DNA reagent | p3036 GST-HPV-16 E6 | Addgene | Plasmid #10849 | |
| Recombinant DNA reagent | V5-TurboID-NES_pCDNA3 | Addgene | Plasmid #107169 | |
| Commercial assay or kit | Dual-Glo Luciferase Assay System | Promega | E2920 | |
| Commercial assay or kit | TnT Coupled Reticulocyte Lysate System | Promega | L4610 | |
| Commercial assay or kit | Trans-Blot Turbo RTA Midi PVDF Transfer Kit | Bio-Rad | #1704275 | |
| Chemical compound, drug | Lipofectamine 2000 transfection reagent | Thermo Fisher Scientific | 11668027 | |
| Chemical compound, drug | Pierce magnetic Streptavidin Beads | Thermo Fisher Scientific | 88817 | |
| Chemical compound, drug | Protein G Dynabeads | Thermo Fisher Scientific | 10003D | |
| Chemical compound, drug | Amersham ECL Prime WB Detection Reagent | Cytiva | RPN2232 | |

## Molecular cloning

For transient protein expression in mammalian cells or in vitro translation of N-terminally Myc-, HA-, or Flag-tagged p53, p63, and p73, PCR-generated inserts were introduced in pcDNA3.Myc, p5RPU. Myc, pcDNA3.HA, or pcDNA3.Flag, respectively, by subcloning using BamHI and XhoI restriction sites. pcDNA3.Myc, pcDNA3.HA, or pcDNA3.Flag are derivatives of pcDNA3.1(+) (Thermo Fisher Scientific) engineered with a Myc, HA-, or Flag-tag between HindIII and BamHI sites. pcDNA3.Myc was further modified by exchanging the CMV promoter with a synthetic minimal promoter 5RPU (*Brown et al., 2017*) yielding p5RPU.Myc. All inserts lack the intrinsic start codon to avoid expression of untagged proteins via alternative translation initiation skipping the respective tag. For the p53 isoform inserts, full-length TAp53α cDNA was used as a template with the isoform-specific β-, γ-, and $\phi$-C-termini sequences being added via the 3'-oligo. The deletion in the Δp53 isoform, as well as any mutations, was generated by site-directed mutagenesis. For the C-terminally Myc-tag p53 isoforms, PCR-generated inserts were subcloned in the pcDNA3.1(+) via BamHI and XhoI restriction sites with a Kozak sequence and start codon being added by the 5'-oligo and the Myc tag by the 3'-oligo.

For the generation of stable cell lines expressing p53 isoform N-terminally fused to Myc-tag and TurboID, PCR-generated p53 inserts were introduced in pTurboN by subcloning using BamHI and XhoI restriction sites. pTurboN was created by inserting a Myc-tag followed by TurboID in pcDNA5/FRT/TO (Thermo Fisher) via the HindIII and BamHI restriction sites. V5-TurboID-NES_pCDNA3 (a gift from Alice Ting (Addgene plasmid #107169; http://n2t.net/addgene:107169; RRID:Addgene_107169)) was used as a template and the Myc-tag was added via the 5'-oligo.

For luciferase reporter assays with the full HSP70 promoter or the HSEs only, the HSP70 promoter (−775–214) or three repeats of the HSP70 promoter HSE (AGAACGTTCTAGAAC) were subcloned in the pGL3-Basic vector (Promega) using NheI and XhoI as restriction sites.

For recombinant expression of the thermostabilized (TS) variants of p53 DBD-OD (41-356), p53 DBD-OD (41-356), p53 DBD (41-312), and Δ40p53beta in *E. coli*, inserts were generated by PCR using a p53 template carrying four thermostabilizing mutations (M133L V203A N239Y N268D) (*Joerger et al., 2006*) and introduced in pET-15b-His$_{10}$-TEV (N-terminal His$_{10}$-tag followed by TEV protease cleavage site MGHHHHHHHHHHHDYDIPTTENLYFQGS), previously described in *Osterburg et al., 2023* by subcloning using BamHI and XhoI restriction sites.

For expression of the HPV16 E6 (AA9–158), an insert was generated by PCR using p3036 GST-HPV-16 E6 (a gift from Peter Howley (Addgene plasmid #10849; http://n2t.net/addgene:10849; RRID:Addgene_10849)) as a template. The insert was subcloned in pGEX-His8-TEV using BamHI and XhoI restriction sites. pGEX-His8-TEV was created by exchanging the 3C protease cleavage site (…GDHPPKSDLEVLFQGPLGS…) in pGEX-6P-2 (Cytvia) with a His8-tag followed by a TEV protease cleavage site (…GDHPPGTHHHHHHHHPGTENLYFQGS…). For the expression of GST only, a stop-codon was inserted in pGEX-His8-TEV using BamHI and XhoI restriction sites.

The cloning of the expression plasmids for the p53 family DBD-specific DARPin G4 and the non-binding control DARPin (cDP) has been previously described (*Strubel et al., 2022*).

## Protein expression and purification

For protein production, *E. coli* BL21(DE3) Rosetta cells (SGC Frankfurt) were transformed with the respective *E. coli* expression plasmid. Cells were grown at 37°C in 2xYT medium to an optical density (OD) of ~0.8 and protein expression was induced with 1 mM IPTG (Carl Roth) for 16–18 hr at 18°C, except for GST only which was expressed at 22°C. For p53 variants and HPV16 E6 the medium was additionally supplemented with 100 µM zinc acetate (Carl Roth). After expression was completed, cells were harvested by centrifugation and the pellet was snap frozen in liquid nitrogen and stored at –80°C until use.

For purification, the cell pellet was thawed on ice and resuspended in immobilized metal affinity chromatography (IMAC) A buffer (25 mM HEPES pH 7.2, 400 mM NaCl, 5% Glycerol, 20 mM β-ME, 10 µM zinc acetate, and 25 mM imidazole) supplemented with DNAse (Sigma-Aldrich), RNAse (Sigma-Aldrich) and self-prepared protease inhibitor. After cell lysis by sonication, the lysate was cleared by centrifugation at 4°C. All proteins were subjected to an initial IMAC purification step. The supernatant was loaded onto a HiTrap IMAC Sepharose FF column (Cytiva) pre-equilibrated in IMAC A buffer, bound proteins were washed with 20 column volumes (CV) IMAC A buffer and eluted with 2 CV IMAC B buffer (IMAC A with 300 mM imidazole). The eluted proteins were then further processed in different ways. All p53 variants were supplemented with self-prepared TEV protease (1:50 wt/wt) for removal of the His-tag and dialysed against IMAC A buffer overnight at 4°C. A subsequent reverse IMAC step was performed to remove TEV protease, the cleaved tag and any uncleaved protein. p53 variants were further purified by heparin-based affinity/ion-exchange (IEX) chromatography. Prior loading on HiTrap Heparin HP columns (Cytiva), the salt concentration of the protein solutions was reduced below 100 mM by dilution with IEX A buffer (25 mM HEPES pH 7.2, 50 mM NaCl, 5% Glycerol, 20 mM β-ME,1 0 µM zinc acetate). Bound protein was eluted with a gradient from 50 mM to 1 M NaCl over 20 CV using IEX B buffer (IEX A with 1 M NaCl). In a final polishing step, p53 proteins were subjected to SEC using a self-packed Superdex 200 SEC column (Cytiva) equilibrated in p53 storage buffer (25 mM HEPES pH 7.5, 150 mM NaCl, 0.5 mM TCEP). Monodisperse peak fractions were pooled, concentrated by centrifugation (Amicon Ultra Centrifugal Filters, Merck KGaA) and snap frozen in liquid nitrogen for storage at –80°C until usage.

GST-HPV16 E6 and GST were further purified by glutathione affinity chromatography (GAC) after IMAC instead. The proteins were loaded onto a GSTrap FF Columns column (Cytiva) pre-equilibrated in IMAC A buffer, bound proteins were washed with 20 CV IMAC A buffer and eluted with 2 CV GAC buffer (IMAC A with 20 mM reduced glutathione instead of imidazole). GST was then subjected to SEC using a self-packed Superdex 75 SEC column (Cytiva) equilibrated in GST storage buffer (25 mM HEPES pH 7.5, 150 mM NaCl, 10% glycerol, 0.5 mM TCEP). Monodisperse peak fractions were pooled, concentrated and stored as described above. As GST-HPV16 E6 aggregates at high concentrations, following GAC the protein was only subjected to a salt exchange using HiTrap Desalting columns with Sephadex G-25 resin (Cytvia) equilibrated in GST storage buffer and directly frozen and stored as described above.

The biotinylated p53 family DBD-specific DARPin G4 and the non-binding control DARPin (cDP) were expressed and purified as previously described (*Strubel et al., 2022*).

## Analytical SEC of purified proteins

Analytical SEC of purified proteins was performed operating a Superdex 10/300 GL with an ÄKTA purifier chromatography system at 4°C. SEC buffer (25 mM HEPES pH 7.5, 150 mM NaCl, and 0.5 mM TCEP) was freshly prepared, filtered and degassed. Proteins were injected on the SEC column

equilibrated in the SEC buffer and elution was monitored via absorbance at 280 nm. For the visualization of analytical SEC results, the absorbance signal was normalized and plotted against the elution volume (Prism 8).

## Analytical SEC of cell lysates

Analytical SEC of cell lysates was performed operating a Superose 6 3.2/300 with an ÄKTA purifier chromatography system at 4°C. The SEC buffer (50 mM TRIS pH 7.5, 150 mM NaCl, 20 mM CHAPS, and 1 mM DTT) was freshly prepared, filtered and degassed. Cells were seeded in 6-well plates and transfected with cell culture expression plasmids encoding the respective proteins. Twenty-four hours after transfection, cells were harvested, resuspended in 120 µl analytical SEC lysis buffer (10 ml SEC buffer supplemented with 2 mM $MgCl_2$ and 1 tablet cOmplete, Mini, EDTA-free), immediately frozen in liquid nitrogen and stored at −80°C until further use. Prior to SEC analysis, cells were thawed in an ice-water bath, supplemented with 2 µl benzonase, incubated for 60 min on ice and centrifuged for 15 min at 16,000 × g and 4°C to remove cell debris. 50 µl supernatant was injected on the SEC column equilibrated in the SEC buffer and 85 µl fractions were collected in a 96-well microtiter plate. For subsequent WB analysis, 40 µl of the fractions were mixed with 10 µl 5x SDS–PAGE sample buffer in 96-well PCR plate and boiled for 5 min at 95°C.

## SPR

Specific DNA binding of recombinantly expressed and purified p53 constructs was measured by SPR spectroscopy using a Biacore X-100 system (Cytiva). The protocol was adopted from *Chen et al., 2011* and has been performed as previously described by us for p53 and p63 (*Osterburg et al., 2023*; *Timofeev et al., 2020*).

Proteins were dialysed overnight at 4°C against SPR buffer (25 mM HEPES pH 7.5, 200 mM NaCl, 0.5 mM TCEP, and 0.005% Tween-20) and centrifuged to remove any aggregates. For the measurements, protein solutions were prepared by serial dilution (factor 2–2.5). Approximately 200 response units (RU) of biotinylated annealed dsDNA with the 20 bp p21 RE was immobilized in flow cell F2 of an SA chip (Cytiva) and an equal amount of the dsDNA with a random sequence in flow cell F1. All measurements were conducted at 20°C and in the multi-cycle format with at least one start-up cycle followed by nine measurement cycles with increasing protein concentrations. Each cycle comprised a 3 min association phase followed by a 2-min dissociation phase. The surface was regenerated by two consecutive 15 s injections of SPR regeneration buffer (25 mM HEPES pH 7.5, 500 mM NaCl, 0.05% SDS). All measurements were performed consecutively with the same chip in three replicates of independently prepared dilution series.

Sensograms were background corrected and affinity binding curves were extracted by equilibrium analysis using the BIAevaluation software (Cytiva). Affinity curves were plotted and fitted with a nonlinear, least squares regression using a single-exponential one-site binding model with Hill slope to determine the dissociation constant $K_D$, the Bmax value and Hill slope factor $h$ (GraphPad Prism 8).

## Thioflavin T assay

Thioflavin T assay was performed using a Spark multimode microplate reader (Tecan) and Thioflavin T to determine aggregational behaviour. Samples were prepared for measurement by mixing 10 µl of peptide stock solution (25 mM HEPES (pH 7.5), 160 mM NaCl, 0.5 mM TCEP, 6 M GndHCl; final peptide concentration of 20 µM) and 0.1 µl Thioflavin T working solution (25 mM HEPES (pH 7.5), 150 mM NaCl, 0.5 mM TCEP; final ThT concentration of 25 µM) to a final volume of 200 µl in black, non-binding polystyrol microplates (Greiner Bio-One). Prior to measuring the amyloid-specific fluorescent signal, the samples were incubated at 37°C for 45 min. Measurements were performed using Ex: 430/20 and Em: 485/20 filters. Fluorescent signals were determined in three independent measurements from the same peptide batch. Statistical significance was assessed by ordinary one-way ANOVA using GraphPad Prism 8.

## Cell culture

The non-small cell lung cancer cell line H1299 (purchased from ATCC) was used because of its p53 deletion and non-detectable or low endogenous levels of p63 and p73 (*Liu et al., 2012*), respectively, resulting in low background levels for the functional assays in this study. Cells were cultured at 37°C

and 5% $CO_2$ in RPMI 1640 medium (Thermo Fisher Scientific) supplemented with 10% fetal bovine serum (FBS, Capricorn Scientific), 100 U/ml penicillin and 100 µg/ml streptomycin (Thermo Fisher Scientific). H1299 cells were routinely tested for mycoplasma contaminations.

For recombinant protein expression, H1299 cells in medium without antibiotics were transfected using Lipofectamine 2000 transfection reagent (Thermo Fisher Scientific) according to the manufacturer's recommendation.

## Generation of stable cell lines

For generation of stable inducible expressing p53 isoform cell lines, the U2OS Flp-In T-Rex cells (gift from C. Behrends, generated according to the protocol of Thermo Fisher Scientific) for homologous recombination of the target genes were used. After 2 weeks of culturing, the T-Rex-U2OS cells were transfected in a 6-well plate using the Lipofectamine 2000 transfection reagent (Thermo Fisher Scientific) with pcDNA5/FRT/TO (Thermo Fisher Scientific) containing p53 family isoforms, as well as pOG44 (Thermo Fisher Scientific) containing the Flp recombinase according to the manufacturer's recommendations. After transfection, DMEM medium containing 10% tetracycline-free FBS (Bio Cell) was used. Twenty-four hours after transfection, cells were reseeded in 15 cm dishes. One day after cell transfer, the medium was exchanged to selection medium (10% tetracycline-free FBS, 4 µg/ml blasticidin, 200 µg/ml hygromycin (Thermo Fisher Scientific), 100 U/ml penicillin, 100 µg/ml streptomycin, and 1 mM pyruvate). Cells were cultured until a non-transfected control showed no viable cells (~10–14 days). Ten single colonies of each cell line were isolated, cultured and inducible expression of the desired proteins was tested via Western blot. Protein expression was induced by adding 1 µg/ml tetracycline (Thermo Fisher Scientific) to the selection medium for 24 hr. For further experiments, three individual clones of each p53 family isoform were chosen.

## Pulldown – mass spectrometry

To prepare proteomic samples for mass spectrometry analysis, stable cell lines expressing p53 variants were induced by adding 1 µg/ml tetracycline to selection medium for 24 hr. After induction, cells were collected and lysed on ice in RIPA lysis buffer (50 mM HEPES (pH 7.5), 150 mM NaCl, 1% Nonidet P-40, 1% sodium deoxycholate, 2 mM $MgCl_2$, 1 mM DTT) with the addition of protease and phosphatase inhibitors. Lysates were sonicated and cleared by centrifugation. Afterwards, 100 µl lysate was taken out from the extracts and used to estimate the protein concentration with the Pierce BCA Protein Assay Kit (Thermo Fisher Scientific). To enrich biotinylated proteins from the protein extracts, 125 µl of streptavidin-coated magnetic beads (Dynabeads MyOne Streptavidin C1, Invitrogen) were washed 5x with wash buffer 1 (50 mM HEPES pH 7.5, 200 mM NaCl, 1 mM DTT, 0.1% Tween-20). Wash buffer 1 was exchanged from the beads with 1 ml protein lysate and incubated on a rotor overnight at 4°C. The beads were sequentially washed 5x with 1 ml wash buffer 1 (50 mM HEPES pH 7.5, 200 mM NaCl, 1 mM DTT, 0.1% Tween-20) followed by five further wash steps with 1 ml wash buffer 2 (50 mM HEPES pH 7.5, 200 mM NaCl). To confirm the successful enrichment of the biotinylated proteins, 75 µl of the suspension was taken out for Western blot analysis and bound proteins were eluted with LDS buffer. The remaining liquid on MS samples was completely removed from the beads. Sample preparation for mass spectrometry was performed through on-bead digestion, employing a buffer containing 2% sodium deoxycholate, 1 mM TCEP and 4 mM CAA, prepared in a 50 mM Tris pH 8.5. The trypsin digestion process was carried out overnight, followed by sample purification using SDB-RPS membranes. Subsequently, the samples underwent a desalting step before being subjected to analysis on the mass spectrometer.

## LC–MS analysis

Samples were analysed on a Q Exactive HF coupled to an easy nLC 1200 (Thermo Fisher Scientific) using a 35-cm long, 75-µm ID fused-silica column packed in house with 1.9 µm C18, and kept at 50°C using an integrated column oven. Peptides were eluted by a non-linear gradient from 4% to 32% acetonitrile (ACN) over 60 min and directly sprayed into the mass spectrometer equipped with a nanoFlex ion source (Thermo Fisher Scientific). Full scan MS spectra (300–1650 $m/z$) were acquired in Profile mode at a resolution of 60,000 at $m/z$ 200, a maximum injection time of 20 ms and an AGC target value of $3 \times 10^6$ charges. Up to 10 most intense peptides per full scan were isolated using a 1.4 Th window and fragmented using higher energy collisional dissociation (normalized collision energy

of 27). MS/MS spectra were acquired in centroid mode with a resolution of 30,000, a maximum injection time of 54 ms and an AGC target value of $1 \times 10^5$. Single charged ions and ions with unassigned charge states were not considered for fragmentation and dynamic exclusion was set to 20 s.

## Mass spectrometry data processing

MS raw data processing was performed with MaxQuant and its in-built label-free quantification algorithm MaxLFQ applying default parameters (*Tyanova et al., 2016*). Acquired spectra were searched against the human reference proteome (Taxonomy ID 9606) downloaded from UniProt (21-11-2018; 94,731 sequences including isoforms) and a collection of common contaminants (244 entries) using the Andromeda search engine integrated in MaxQuant (*Cox et al., 2011*). Identifications were filtered to obtain false discovery rates (FDR) below 1% for both peptide spectrum matches (minimum length of 7 amino acids) and proteins using a target-decoy strategy (*Elias and Gygi, 2007*). Further statistical analysis was performed on Perseus software.

## Sample processing for LC–MS/MS with a focus on p53-specific peptides

Eluted proteins (30 µg) were prepared for bottom-up proteomics using a suspension trapping protocol. Briefly, samples were mixed with lysis buffer (10% SDS, 100 mM tetraethylammonium bicarbonate (TEAB), pH 7.55, with $H_3PO_4$) in a 1:1 ratio, reduced using 20 mM dithiothreitol for 10 min at RT, and alkylated with 50 mM iodoacetamide for 30 min at RT in the dark. Subsequently, the samples were acidified using phosphoric acid in a final concentration of ~1.2%. Binding/wash buffer (BW: 90% methanol, 50 mM TEAB, pH 7.5 with $H_3PO_4$) was added in a 1:7 ratio. The protein suspension was loaded onto the S-trap filtered (size: 'micro'; ProtiFi) by centrifugation for 20 s at 4000 × *g*. Trapped proteins were washed with 150 µl of BW buffer four times. Trypsin (1 µg; Promega) was added in 60 µl of 40 mM ammonium bicarbonate buffer. Digestion was performed overnight (~18 hr) at RT in a humidified chamber. Peptides were collected by washing in three consecutive steps by centrifugation at 4000 × *g* for 40 s starting with digestion buffer and two washes of 0.2% formic acid in MS-grade water ($H_2O$). Peptides were desalted using C18 SPE cartridges (25 mg, INSOLUTE EC, Biotage) and dried in vacuo at 45°C.

## LC–MS/MS analysis of turboID experiment

Dried peptides were reconstituted in 20 µl reconstitution buffer (95% $H_2O$, 5% ACN with 0.1% FA) and analysed using a nanoElute 2 nano-HPLC coupled to a timsTOF HT mass spectrometer via a captive spray ion source (1600 V, Bruker Daltonics). Peptides were loaded directly onto the analytical column (15 cm × 150 µm column with 1.5 µm C18-beads (PepSep); at max. 800 bar) maintained at 60°C and connected to a 20-µm ZDV sprayer (Bruker Daltonics). In 25-min runs, peptides were separated in a linear gradient of water (buffer A: 100% $H_2O$ and 0.1% FA) and ACN (buffer B: 100% ACN and 0.1% FA) ramping from 2% to 38% B in 21 min with a constant flow rate of 800 nl/min, followed by a wash of 95% B for 4 min with a flow rate of 1 µl/min. For data acquisition in DIA mode, the 'short-gradient' dia-PASEF method was employed. In brief, 21 dia-PASEF windows were distributed to a TIMS scan each, covering a *m/z* range from 475 to 1000 *m/z* in 25 Da DIA windows, resulting in an estimated cycle time of 0.95 s. The ion mobility range was covered from 1.30 to 0.85 Vs/cm². For further details, all method parameters are embedded in the uploaded raw data.

## Processing of DIA LC–MS/MS data of turboID experiment

DIA-MS raw files were analysed with the open-source software DIA-NN (version 1.9; 'PMID: 31768060') using a library-free approach. The predicted library was generated using the in silico FASTA digest (Trypsin/P) option with the human UniProtKB database (swissprot, tremble, download: 08.05.2023). Common MS contaminants were included in the search (Cambridge Centre for Proteomics (CCP) database). Deep learning-based spectra and RT prediction was enabled. The observed peptide length range was set to 7–35 amino acids, missed cleavages to 2 and precursor charge range to 1–5. N-terminal methionine excision, methionine oxidation, and N-terminal acetylation were set as variable modification, cysteine carbamidomethylation as a fixed modification. The number of variable modifications per peptide was limited to 3. MS1 and MS2 mass accuracies were set to 15 ppm. Precursor and run-specific protein group FDRs were filtered for hits ≤1%. Scan windows were set to 0, isotopologues were enabled, while the shared spectra option,

match-between-runs, and LFQ normalization were disabled. Protein inference was performed using genes with the heuristic protein inference option enabled. The neural network classifier was set to single-pass mode and the selected quantification strategy was 'QuantUMS (high precision)'. The cross-run normalization was set to 'RT-dependent', the library generation to 'smart profiling', the speed and RAM usage to 'optimal results'. To evaluate protein-level LFQ data, the DIA-NN report table was imported into the statistical computing software R. Protein and precursor output tables were analysed in the statistical computing software R, filtering for proteotypicity and discarding contaminant hits. Using prinicipal component analysis and unsupervised hierarchical clustering of Pearson correlation coefficients, four outlier samples were identified in an unbiased manner and thus removed from downstream analysis. All visualizations were performed using ggplot2 of the tidyverse R package.

## Luciferase reporter assay

Transcriptional activity was analysed via a reporter assay. H1299 cells were transfected with pRL-CMV (Promega) in combination with pBV-Luc BDS-2 3x WT (pBDS-2) (BDS-2 3x WT (p53-binding site) was a gift from Bert Vogelstein (Addgene plasmid #16515; http://n2t.net/addgene:16515; RRID:Addgene_16515)), pGL2 with Mdm2 promoter, pGL3 with p21 promoter (**Straub et al., 2010**), pGL3 with the HSP70 promoter or pGL3 with HSP70/3x HSEs, and pcDNA3.1(+) as an empty vector control or pcDNA3.Myc plasmids encoding the indicated Myc-tagged p53, p63, or p73 variants.

Cells were transfected 1 day after seeding. Twenty-four hours after transfection, cells were harvested and resuspended in 450 µl fresh medium. Per sample, 45 µl cell suspension was transferred in four wells each of a white Nunc 96-well microplates (Thermo Fisher Scientific) to determine the luciferase signal in technical quadruplicates.

Activities of Firefly and Renilla luciferase were measured with the Dual-Glo Luciferase Assay System (Promega) according to the manufacturers' recommendation. Luminescence signals were measured using a Spark multimode or a GENios Pro microplate reader (Tecan). Analysis was performed by calculation of the ratio of luminescence from the experimental reporter (Firefly) to luminescence from the control reporter (Renilla) for each technical replicate and normalized to the empty vector control to yield the relative fold induction for each biological replicate. For further input sample preparation for Western blot analysis, the residual cells were centrifuged, resuspended in 150 µl 2x SDS–PAGE sample buffer and boiled.

## SDS–PAGE and Western blotting

Proteins were separated by SDS–PAGE using Mini-PROTEAN Tetra Cell SDS-PAGE system (Bio-Rad) in combination 4–15% Mini-PROTEAN TGX Stain-Free Precast Protein gels (Bio-Rad). The SDS–PAGE gels were blotted onto PVDF membranes using the Trans-Blot Turbo Transfer System in combination with the Trans-Blot Turbo RTA Midi PVDF Transfer Kit from Bio-Rad and the SDS–PAGE Transfer Buffer according to the manufacturer's recommendation. Membranes were incubated with blocking solution (5% skim milk powder (Sigma-Aldrich) in TBSt (TBS supplemented with 0.05% (vol/vol) Tween 20)) for 1 hr at room temperature under shaking conditions. Subsequently, the membranes were incubated with the first antibody diluted in blocking solution overnight at 4°C. Afterwards, the membrane was washed three times with TBS-t and incubated with the appropriate peroxidase-conjugated secondary antibody (Jackson ImmunoResearch) in blocking solution and washed three times again. The chemiluminescence signal was detected with a Lumi Imager F1 documentation system using Amersham ECL Prime WB Detection Reagent (Cytiva). Densitometric analysis of Western blots was performed using ImageJ (Version 1.51) and/or using Bio-Rad ChemiDoc Image Lab 6.1. Statistical significance was assessed by ordinary one-way ANOVA followed by Tukey's post hoc test using GraphPad Prism 8.

The following primary antibodies were used in this study: anti-Myc (clone 4A6, Merck KGaA, 1:1000), anti-HA (goat polyclonal A190-107A, Bethyl Laboratories, 1:1000), anti-p53 (DO-1, Santa Cruz Biotechnologies, 1:1000), anti-Vinculin (clone 7F9, Santa Cruz Biotechnology, 1:1000), anti-GST-HRP (RPN1236, GE Healthcare Life Sciences, 1:10,000), anti-Streptavidin-HRP (S2436 Sigma-Aldrich Chemie GmbH, 1:10,000), anti-Flag (M2 clone, Sigma-Aldrich Chemie GmbH, 1:1000), and anti-HSP70/HSC70 (sc-24 clone, Santa Cruz Biotechnology, 1:1000).

## Blue Native–PAGE

Blue Native (BN)–PAGE analysis to assess the oligomeric state and aggregation of p53 variants was performed as described previously (*Kehrloesser et al., 2016*). H1299 cells were harvested 24 hr after transfection and lysed for 30 min at room temperature in 100 µl BN-PAGE lysis buffer (25 mM TRIS-HCl pH 7.5, 150 mM NaCl, 20 mM CHAPS, 1 mM DTT and 2 mM $MgCl_2$) supplemented with 1x protease inhibitor (Roche) and 1 µl benzonase (Merck). Samples were supplemented with 3x BN–PAGE solution (60% glycerol and 15 mM Coomassie Brilliant Blue G-250) and separated for 1 hr at 150 V, followed by 1 hr at 250 V, using the NativePAGE Bis-Tris Gel System with 3–12% Bis-Tris Mini Protein Gels (Thermo Fisher Scientific) at 4°C. Subsequent immunoblotting and detection was performed using the XCell II blot system (Thermo Fisher Scientific) as described above with the exception that membranes were destained with methanol and fixed with 8% acetic acid before blocking. For parallel SDS–PAGE analysis, lysates were supplemented with 5x SDS–PAGE sample buffer (250 mM TRIS pH 8.0, 7.5% (wt/vol) SDS, 25% (vol/vol) glycerol, 0.025% (wt/vol) bromphenol blue sodium salt, and 12.5% (vol/vol) β-ME), boiled and subjected to SDS–PAGE followed by immunoblotting as described above.

## DNA pulldown assay

Per pulldown 50 pmol of biotinylated annealed p21 RE dsDNA oligonucleotide were immobilized on 10 µl MyOne T1 Dynabeads (Thermo Fisher) in DNA pulldown buffer (50 mM HEPES pH7.5; 200 mM NaCl; 0.1% Tween-20; 0.5 mM TCEP) for 1 hr at 4°C. Myc-tagged p53 isoforms were produced by in vitro translation (#L1170, Promega) from the respective pcDNA3.1(+) plasmids. Expression was conducted for 90 min at 30°C and the lysate was cleared afterwards by centrifugation. As input samples, 5 µl of the expressions were mixed with 95 µl 2x SDS–PAGE sample buffer and boiled. Beads were washed three times with DNA pulldown buffer to remove unbound DNA and incubate with 20 µl in vitro translated p63 in DNA pulldown buffer with a final volume of 400 µl for 3 hr at 4°C rotating. Afterwards, beads were washed four times with 400 µl DNA pulldown buffer and bound proteins were eluted by incubation of the beads in 100 µl 1x LDS buffer supplemented with reducing agent for 10 min at 70°C. Input and pulldown samples were analysed by Western blotting and quantified. For the relative pulldown efficiency, each pulldown sample was normalized to the input sample. The DNA pulldown assays were performed as three independent biological replicates. Statistical significance was assessed by ordinary one-way ANOVA followed by Tukey's post hoc test using GraphPad Prism 8.

## Solubility assay

To further investigate the aggregation state of the p53 mutants and isoforms within the cells, a solubility assay was performed. Cells were transfected 1 day after seeding in 12-well plates. Twenty-four hours after transfection, cells were washed with 1 ml PBS, harvested with accutase and spun down. The supernatant was aspirated, and the cell pellet was washed again with PBS followed by centrifugation. Afterwards, the cells were lysed with 100 µl solubility buffer A (20 mM Tris (pH 7.5), 150 mM NaCl, 1 mM DTT, 2 mM $MgCl_2$, 1% Triton X-100) with 1 µl benzonase for 60 min on ice. Followed by centrifuging for 15 min at 13,000 rpm at 4°C. 75 µl of the supernatant (containing the soluble fraction) was mixed with 25 µl 5x SDS sample buffer. The remaining supernatant was discharged and the pellet (containing the insoluble fraction) was resuspended in 100 µl in solubility buffer B (20 mM Tris (pH 7.5), 150 mM NaCl, 1 mM DTT, 2 mM $MgCl_2$, 1% SDS). The samples were incubated for 20 min at room temperature. After incubation, the samples were spun down for 15 min at 13,000 rpm at room temperature. 75 µl of the supernatant was mixed with 25 µl 5x SDS sample buffer. The samples were incubated at 95°C for 10 min. Protein levels were analysed by Western blot. The solubility assays were performed as three independent biological replicates. Statistical significance was assessed by ordinary one-way ANOVA followed by Tukey's post hoc test using GraphPad Prism 8. To investigate the effect of chaperones on the solubility of p53 isoforms, cells were transfected and treated with different inhibitors 1 day after seeding in 12-well plates. Twenty-four hours after transfection, the solubility assay was performed as described above. The used inhibitors were JG-98 (HSP70 inhibitor, final concentration 5 µM, MedChem Express), YM-1 (HSP70 inhibitor, final concentration 10 µM, MedChem Express), VER-155008 (HSP70 inhibitor, final concentration 10 µM, MedChem Express), and 17-AAG (HSP90 inhibitor, final concentration 2 µM, MedChem Express).

## Reticulocyte lysate protein expression

Proteins were translated in vitro using the TnT Coupled Reticulocyte Lysate System (Promega). Constructs in a pcDNA3.1(+) vector were diluted to 100 ng/µl and mixed in a 1:4 volume ratio with RRL and incubated for 90 min at 30°C. The reaction was stopped by adding Benzonase (Millipore) for 30 min at 4°C. The supernatant was cleared by centrifugation at $16,100 \times g$ for 10 min at 4°C and stored on ice until use.

## DARPin pulldown assays

Target proteins were in vitro translated. An excess of biotinylated DARPins was pre-incubated with pre-equilibrated magnetic Pierce Streptavidin Beads (Thermo Fisher Scientific) in Pulldown (PD) wash buffer (50 mM HEPES, pH 7.5, 200 mM NaCl, 0.1% (vol/vol) Tween-20, 0.5 mM TCEP) while rotating for 1 hr at 4°C. The magnetic beads were washed three times with PD wash buffer to remove unbound DARPins and were resuspended in PD wash buffer with the same volume as before to maintain magnetic bead concentrations. 10 µl DARPin loaded beads were mixed with 10 µl in vitro translated protein and adjusted to a total volume of 400 µl with PD wash buffer. The PD mix was incubated while rotating for 3 hr at 4°C. Pulldown samples were washed five times with 400 µl PD wash buffer and eluted with 1x LDS buffer (Thermo Fisher Scientific, supplemented with reducing agent) by boiling at 70°C for 10 min. Samples were analysed by Western blot. The DARPin pulldown assays were performed as three independent biological replicates. Statistical significance was assessed by ordinary one-way ANOVA followed by Tukey's post hoc test using GraphPad Prism 8.

## HPV E6 degradation assay

The E6 protein from HPV is able to induce p53 hyper-degradation, consequently inactivating p53 when the DBD of p53 is correctly folded (*Scheffner et al., 1990*). N-terminally Myc-tagged proteins were in vitro translated using RRL. Lysates were diluted in reaction buffer (25 mM HEPES pH 7.5, 150 mM NaCl, 0.5 mM TCEP, 1 mM $MgCl_2$, 1 mM ATP) and supplemented with either 5 µM GST-tagged HPV16 E6 or GST only as control. Reactions were incubated for 4 hr at 25°C. The reaction was stopped with addition of 30 µl 2x SDS sample buffer and incubation for 3 min at 95°C. Protein levels were analysed by Western blot. The HPV E6 assays were performed as three independent biological replicates. Statistical significance was assessed by ordinary one-way ANOVA followed by Tukey's post hoc test using GraphPad Prism 8.

## Co-immunoprecipitation

For co-IP, H1299 cells were co-transfected with either HA-tagged wtp53, p63, or p73 and different isoforms of Myc-tagged p53. For co-IP with endogenous HSC/HSP70, H1299 cells were transfected with Myc-tagged p53 variants alone. Twenty-four hours after transfection, cells were collected and lysed on ice in RIPA lysis buffer (50 mM HEPES (pH 7.5), 150 mM NaCl, 1% Nonidet P-40, 1% sodium deoxycholate, 2 mM $MgCl_2$, 1 mM DTT) with the addition of protease and phosphatase inhibitors. Lysates were cleared by centrifugation and incubated with 0.5 µg of HA-antibody or HSC/HSP70-antibody overnight at 4°C. The immunocomplexes were removed from the lysate using 12 µl Protein G Dynabeads (Thermo Fisher Scientific), washed four times with ice-cold IP wash buffer (50 mM Tris (pH 7.8), 150 mM NaCl, 0.1% Tween-20) and eluted with 1x LDS-sample buffer (Thermo Fisher Scientific) supplemented with reducing agent (Thermo Fisher Scientific) for 10 min at 70°C. Samples were analysed by Western blotting. Relative immunoprecipitation efficiency was calculated by normalization of the IP Western blot signal to the respective input signal.

## Conformation-specific immunoprecipitation

For conformation-specific immunoprecipitation (Conf-IP), H1299 cells were transfected with empty vector or N-terminally Myc-tagged p53 variants. p53 isoforms were IP with either α-Myc or α-p53 antibody (pAB240). The latter binds an epitope in the DBD of p53, which is only exposed when the domain is unfolded (*Schmieg and Simmons, 1993*). Twenty-four hours after transfection, cells were collected and lysed on ice in RIPA lysis buffer (50 mM HEPES (pH 7.5), 150 mM NaCl, 1% Nonidet P-40, 1% sodium deoxycholate, 2 mM $MgCl_2$, 1 mM DTT) with the addition of protease and phosphatase inhibitors. Lysates were cleared by centrifugation and incubated with 0.5 µg of α-Myc or pAB240 antibody overnight at 4°C. The immunocomplexes were removed from the lysate using 12 µl Protein

G Dynabeads (Thermo Fisher Scientific), washed four times with ice-cold IP wash buffer (50 mM Tris (pH 7.8), 150 mM NaCl, 0.1% Tween-20) and eluted with 1x LDS-sample buffer (Thermo Fisher Scientific) supplemented with reducing agent (Thermo Fisher Scientific) for 10 min at 70°C. Samples were analysed by Western blotting. Relative immunoprecipitation efficiency was calculated by normalization of the IP Western blot signal to the respective input signal.

## Statistical analysis

If not stated otherwise, experiments were performed as three independent biological replicates followed by statistical analysis (SD; $n = 3$) and statistical significance was assessed by ordinary one-way ANOVA (n.s.: $p > 0.05$, *$p \leq 0.05$, **$p \leq 0.01$, ***$p \leq 0.001$, ****$p \leq 0.0001$) followed by Tukey's post hoc test using GraphPad Prism 8.

## Material availability

Cell lines stably expressing the individual p53 isoforms created during this research project are available upon request.

## Acknowledgements

We thank Marina Hoffmann and Thorsten Mosler for help with deposition of the mass spectrometry data to the PRIDE server. Funding was provided by the Deutsche Krebshilfe (Grant Number 70113956), the Centre for Biomolecular Magnetic Resonance (BMRZ), and the Clusterproject ENABLE (funded by the Hessian Ministry for Science and the Arts).

## Additional information

### Competing interests

Volker Dötsch: Senior editor, eLife. The other authors declare that no competing interests exist.

### Funding

| Funder | Grant reference number | Author |
|---|---|---|
| Deutsche Krebshilfe | 70113956 | Ivan Dikic Volker Dötsch |
| Hessian Ministry of Arts for Science and the Arts | Clusterproject ENABLE | Ivan Dikic Volker Dötsch |

The funders had no role in study design, data collection, and interpretation, or the decision to submit the work for publication.

### Author contributions

Anamari Brdar, Formal analysis, Investigation, Visualization, Writing - original draft; Christian Osterburg, Conceptualization, Formal analysis, Supervision, Investigation, Visualization, Writing – review and editing; Philipp Münick, Formal analysis, Investigation, Visualization, Writing – review and editing; Anne Christin Machel, Birgit Schäfer, Investigation; Rajeshwari Rathore, Formal analysis, Visualization; Susanne Osterburg, Formal analysis, Investigation; Büşra Yüksel, Data curation, Formal analysis; Kristina Desch, Data curation, Formal analysis, Investigation; Julian D Langer, Conceptualization, Data curation, Formal analysis, Supervision, Writing – review and editing; Ivan Dikic, Volker Dötsch, Conceptualization, Supervision, Funding acquisition, Project administration, Writing – review and editing

### Author ORCIDs

Philipp Münick https://orcid.org/0000-0002-0548-0897
Julian D Langer https://orcid.org/0000-0002-5190-577X
Ivan Dikic https://orcid.org/0000-0001-8156-9511
Volker Dötsch https://orcid.org/0000-0001-5720-212X

Reviewer #1 (Public review): https://doi.org/10.7554/eLife.103537.3.sa1

Author response https://doi.org/10.7554/eLife.103537.3.sa2

## Additional files

### Supplementary files
MDAR checklist

Supplementary file 1. Core protein hits for p53 isoforms associated with various chaperones in-vivo.

Source code 1. Code used to analyse the proteomics data.

### Data availability
The mass spectrometry proteomics data have been deposited to the ProteomeXchange Consortium via the PRIDE (*Perez-Riverol et al., 2022*) partner repository with the dataset identifier PXD045655. The data from the second, p53 peptide-specific mass spectrometry experiment have been deposited with the dataset identifier PXD058245. Scripts used to analyse the proteomics datasets are uploaded as *Source code 1*.

The following datasets were generated:

| Author(s) | Year | Dataset title | Dataset URL | Database and Identifier |
|---|---|---|---|---|
| Hoffmann M | 2025 | p53 isoforms have a high aggregation propensity, interact with chaperones and lack binding to p53 interaction partners | https://www.ebi.ac.uk/pride/archive/projects/PXD045655 | PRIDE, PXD045655 |
| Langer J | 2025 | p53 isoforms have a high aggregation propensity, interact with chaperones and lack binding to p53 interaction partners | https://www.ebi.ac.uk/pride/archive/projects/PXD058245 | PRIDE, PXD058245 |

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
