## [Editor Report · eLife Assessment]

This manuscript provides an **important** biochemical analysis of p53 isoforms, highlighting their aggregation propensity, interaction with chaperones, and dominant-negative effects on p53 family members. The authors have substantially strengthened the original manuscript by incorporating new mass spectrometry data and clarifying isoform-specific oligomerization behavior. Although the use of high expression levels limits direct physiological interpretation, the work is carefully framed as an investigation of protein misfolding and stability. Overall, this study offers **convincing** insights into p53 isoform biophysics with broad implications for cancer biology.

---

## [Referee Report · Reviewer #1 (Public review)]

Summary:

Brdar, Osterburg, Munick, et al. present an interesting cellular and biochemical investigation of different p53 isoforms. The authors investigate the impact of different isoforms on the in-vivo transcriptional activity, protein stability, induction of the stress response, and hetero-oligomerization with WT p53. The results are logically presented and clearly explained. Indeed, the large volume of data on different p53 isoforms will provide a rich resource for researchers in the field to begin to understand the biochemical effects of different truncations or sequence alterations.

Strengths:

The authors achieved their aims to better understand the impact/activity of different p53 is-forms, and their data well support their statements. Indeed, the major strengths of the paper lie in its comprehensive characterization of different p53 isoforms and the different assays that are measured. Notably, this includes p53 transcriptional activity, protein degradation, induction of the chaperone machinery, and hetero-oligomerization with wtp53. This will provide a valuable dataset where p53 researchers can evaluate the biological impact of different isoforms in different cell lines. The authors went to great lengths to control and test for the effect of (1) p53 expression level, (2) promotor type, and (3) cell type. I applaud their careful experiments in this regard.

Comments on revised version:

The authors have addressed all of my concerns convincingly, including with a new mass spectrometry experiment to quantify p53 peptides specifically.

---

## [Author Response]

The following is the authors’ response to the original reviews

**Public Reviews:**

**Reviewer #1 (Public review):**
Summary:Brdar, Osterburg, Munick, et al. present an interesting cellular and biochemical investigation of different p53 isoforms. The authors investigate the impact of different isoforms on the in-vivo transcriptional activity, protein stability, induction of the stress response, and hetero-oligomerization with WT p53. The results are logically presented and clearly explained. Indeed, the large volume of data on different p53 isoforms will provide a rich resource for researchers in the field to begin to understand the biochemical effects of different truncations or sequence alterations.Strengths:The authors achieved their aims to better understand the impact/activity of different p53 is-forms, and their data will support their statements. Indeed, the major strengths of the paper lie in its comprehensive characterization of different p53 isoforms and the different assays that are measured. Notably, this includes p53 transcriptional activity, protein degradation, induction of the chaperone machinery, and hetero-oligomerization with wtp53. This will provide a valuable dataset where p53 researchers can evaluate the biological impact of different isoforms in different cell lines. The authors went to great lengths to control and test for the effect of (1) p53 expression level, (2) promotor type, and (3) cell type. I applaud their careful experiments in this regard.Weaknesses:One thing that I would have liked to see more of is the quantification of the various pull-down/gel assays - to better quantify the effect of, e.g., hetero-oligomerization among the various isoforms. In addition, a discussion about the role of isoforms that contain truncations in the IDRs is not available. It is well known that these regions function in an auto-inhibitory manner (e.g. work by Wright/Dyson) and also mediate many PPIs, which likely have functional roles in vivo (e.g. recruiting p53 to various complexes). The discussion could be strengthened by focusing on some of these aspects of p53 as well.

Thank you for these comments. In this paper we have focused on the importance of the integrity of the folded domains of p53 for their function. The unfolded regions in the N- and the C-terminus have not been our main target but the reviewer is right that they play important regulatory functions that are lost in the corresponding isoforms. We have, therefore, added a few sentences in the Discussion section.

With respect to a better quantification, we have re-evaluated the quantification and adjusted where necessary (see also reviewer 2). With respect to the hetero-oligomerization we have run a new mass spectrometry experiment in which we only focus on the p53 peptides. These have been now quantitatively evaluated and the results are provided in this manuscript Fig. 5.

**Reviewer #2 (Public review):**
Summary:In this manuscript entitled "p53 isoforms have a high aggregation propensity, interact with chaperones and lack 1 binding to p53 interaction partners", the authors suggest that the p53 isoforms have high aggregation propensity and that they can co-aggregate with canonical p53 (FLp53), p63 and p73 thus exerting a dominant-negative effect.Strengths:Overall, the paper is interesting as it provides some characterization of most p53 isoforms DNA binding (when expressed alone), folding structure, and interaction with chaperones. The data presented support their conclusion and bring interesting mechanistic insight into how p53 isoforms may exert some of their activity or how they may be regulated when they are expressed in excess.Weaknesses:The main limitation of this manuscript is that the isoforms are highly over-expressed throughout the manuscript, although the authors acknowledge that the level of expression is a major factor in the aggregation phenomenon and "that aggregation will only become a problem if the expression level surpasses a certain threshold level" (lines 273-274 and results shown in Figures S3D, 6E). The p53 isoforms are physiologically expressed in most normal human cell types at relatively low levels which makes me wonder about the physiological relevance of this phenomenon.Furthermore, it was previously reported that some isoforms clearly induce transcription of target genes which are not observed here. For example, p53β induces p21 expression (Fujita K. et al. p53 isoforms Delta133p53 and p53beta are endogenous regulators of replicative cellular senescence. Nat Cell Biol. 2009 Sep;11(9):1135-42), and Δ133p53α induces RAD51, RAD52, LIG4, SENS1 and SOD1 expression (Gong, L. et al. p53 isoform D113p53/D133p53 promotes DNA double-strand break repair to protect cell from death and senescence in response to DNA damage. Cell Res. 2015, 25, 351-369. / Gong, L. et al. p53 isoform D133p53 promotes the efficiency of induced pluripotent stem cells and ensures genomic integrity during reprogramming. Sci. Rep. 2016, 6, 37281. / Horikawa, I. et al. D133p53 represses p53-inducible senescence genes and enhances the generation of human induced pluripotent stem cells. Cell Death Differ. 2017, 24, 1017-1028. / Gong, L. p53 coordinates with D133p53 isoform to promote cell survival under low-level oxidative stress. J. Mol. Cell Biol. 2016, 8, 88-90. / Joruiz et al. Distinct functions of wild-type and R273H mutant Δ133p53α differentially regulate glioblastoma aggressiveness and therapy-induced senescence. Cell Death Dis. 2024 Jun 27;15(6):454.) which demonstrates that some isoforms can induce target genes transcription and have defined normal functions (e.g. Cellular senescence or DNA repair).However, in this manuscript, the authors conclude that isoforms are "largely unfolded and not capable of fulfilling a normal cellular function" (line 438), that they do not have "well defined physiological roles" (line 456), and that they only "have the potential to inactivate members of the p53 protein family by forming inactive hetero complexes with wtp53" (line 457-458).

Therefore, I think it is essential that the authors better discuss this major discrepancy between their study and previously published research.

This manuscript is not about hunting for the next “signal transduction pathway” that is “regulated” by a specific p53 isoform. For such a project work has indeed to be conducted at the endogenous level. However, our manuscript is about the basic thermodynamic behavior of these isoforms in in vitro assays and in some cell culture assays.

What, however, depends on the expression level is the interaction with chaperones as well as the tendency to aggregate. And this we actually show in our manuscript by using two different promotors with very different strength: Strong overexpression leads to aggregation, much weaker expression to soluble isoforms. For the mass spectrometry experiments we have established stable expressing cell lines and not used transiently overexpressing ones.

The level from which on the chaperone systems of the cell cannot keep these isoforms soluble and they start to aggregate is certainly an important question, and we have experimental evidence that if we use different chaperone inhibitors the percentage of the aggregating isoforms in the insoluble fraction increases.

Proteins have to follow the basic physicochemical rules also in cells. And this manuscript sets the stage for re-interpreting the observed cellular effects – not in terms of specific interaction with certain promoters but as causing a stress response and non-specific interaction with other not-well folded domains of other proteins.

With respect to this discussion about the physiological relevance, it is interesting to look at a study that was published in Cell:

Rohaly, G., Chemnitz, J., Dehde, S., Nunez, A.M., Heukeshoven, J., Deppert, W. and Dornreiter, I. (2005) A novel human p53 isoform is an essential element of the ATR-intra-S phase checkpoint. Cell, 122, 21-32.

This manuscript describes how a specific isoform regulates an important pathway. Two other studies also focused on the same isoform but showed that it lacks the nuclear localization signal and therefore does not enter the nucleus. And even if it would, it would have no transcriptional activity due to the unfolding of the DBD.

Chan, W.M. and Poon, R.Y. (2007) The p53 Isoform Deltap53 lacks intrinsic transcriptional activity and reveals the critical role of nuclear import in dominant-negative activity. Cancer Res, 67, 1959-1969.

Garcia-Alai, M.M., Tidow, H., Natan, E., Townsley, F.M., Veprintsev, D.B. and Fersht, A.R. (2008) The novel p53 isoform "delta p53" is a misfolded protein and does not bind the p21 promoter site. Protein Sci, 17, 1671-1678.

This example shows that it is important to re-consider the basic principles of protein structure and protein folding. And that is exactly what this manuscript is about.

**Recommendations for the authors:**

**Reviewer #1 (Recommendations for the authors):**
(1) Does the p53g C-terminus (322-346) form cross-beta amyloid structures? The strong fluorescence signal in the presence of ThT suggests this may be forming amyloid. I wonder if any amyloid sequence predictors identify this region as amyloidogenic.

Using the Waltz predictor (https://doi.org/10.1038/nmeth.1432), the amino acids 339-346 have been identified as potentially amyloidogenic. We have added this information to the manuscript.

(2) The chaperone binding results in Figure 5 are interesting and indeed suggest that many p53 isoforms interact with chaperones in vivo to counteract their destabilized nature. For the 5 p53 isoforms shown in Figure 5D, do they present any HSP70-binding motifs that may not exist in wtp53? These motifs can be predicted from the sequence with established software in a similar manner as the authors performed for TANGO.

**Author response image 1. sa2fig1:** Predicted Chaperon binding sites using the LIMBO prediction tool. (http://www.ncbi.nlm.nih.gov/pubmed/19696878)

We have analyzed the sequence of p53 and the isoforms for potential HSP70 binding sites using the LIMBO prediction tool. The results are shown in the figure above. Wild type p53 has a very strong site that is lost in the β- and ɣ-isoforms. The ɣ-isoform in addition loses another predicted binding site which is replaced with a ɣ-specific one. Overall, this analysis does not provide a very clear picture due to the loss of some and the creation of new, isoform-specific binding sites. We have, therefore, not included this analysis in the manuscript but show it here for the reviewers.

(3) The mixed hetero-tetramers detected by the MS is very interesting. Also the pull-down experiments in Figure 6. However, the extent of hetero-oligomerization is at times hard to follow. Could you more clearly summarize and/or quantify the results of the hetero-oligomerization experiments?

We have conducted a new mass spectrometry experiment that was focused only on the analysis of p53 peptides. These data are now shown in Figure 5 and Supplementary Figure 6. They show that peptides not present in the Δ133p53α isoform and therefore must come from wild type p53 can be detected. For the Δ133p53β isoform these peptides are absent, suggesting that this isoform does not hetero-oligomerize with wild type p53. Furthermore, all β- and ɣ- isoforms do not show peptides derived from wild type p53, again suggesting that they cannot hetero-oligomerize due to the lack of a functional oligomerization domain.

(4) There is a typo in Figure 5. The figure title (top of page) says "Figure 4: Chaperons". Also, "chaperons" appears in the legend.

Thank you for making us aware of this problem. This has been corrected.

(5) The figures are often quite small with a lot of white space. Figure 4 in particular is arranged in a confusing way with A, D, B, C, E, F, G in T->B L->R order. Perhaps some figures could be expanded or re-arranged to make better use of the available space. E.g. could move B, C above panel D, and then shift F, G to be next to E. This would give you A, [B, C, D], [E, F, G] in a 2x2 format.

We have rearranged figures 2, 4, 5 and 6 to be able to enlarge the individual figure panels.

**Reviewer #2 (Recommendations for the authors):**
(1) Figure 2C: Why is the p21-Luc reporter assay performed in SAOS-2 cells when all other assays are performed in H1299?

The assays we have performed in this study are independent of the cell type because we investigate very basic principles of protein folding and stability. If one removes a third of a folded domain, this domain will no longer fold, independent of the cell type it is in. However, to show, that the cell type indeed does not play any role, we have repeated the experiments in H1299 cells. These data are now shown in Figure 2C and the original data in SAOS cells we have moved to Supplementary Figure 1E.

(2) Figure 3: I find the statistics on this figure very confusing... It looks like every isoform is compared to the "WT", but in that case, in Figure 3B for example, how can the Δ40p53β be ****, Δ133p53γ be *** while the Δ133p53α, more different to WT and narrower error bars is non-specific? I guess this comes from the normalization of the GST expression of each isoform but in this case, the isoforms should not be compared to the WT, but to their respective GST sample.

There was indeed a mistake in the statistics, thank you for pointing this out.

We repeated the statistical analysis and the relative protein level within each sample is now calculated using the ratio between the respective GST sample and the sample containing E6. Significance for each isoform was assessed by comparing the relative protein level to the protein level of the WT.

(3) Figures 3D and 3E: the authors did not perform the assays on Δ40p53 isoforms because they "contain a fully folded DBD" (lines 218-219). This may be true for Δ40p53α as shown by the pAB240 binding figure 3C, but it is speculative for Δ40p53β and Δ40p53γ since these were not tested in Figure 3C either... Furthermore, Figure 3B suggests that there may be differences between Δ40p53α, Δ40p53β and Δ40p53γ and therefore these two isoforms should be tested for pAB240 IP at least (and DARPin as well if the pAB240 IP shows differences). Also, why were the TAp53β and TAp53γ not tested in Figures 3D and 3E?

Here we disagree with the reviewer. The PDB is full of structures of the p53 DNA binding domain. All of them – including many structures of the same domain from other species – span residues ~90 to 294 (or the equivalent residues in other species). That means that the β- and ɣ- versions of p53 contain the full DNA binding domain. In contrast to the DNA binding domain, the oligomerization domain, however, is truncated and therefore does not form functional tetramers. This is the reason for the reduced binding affinity to DNA.

The pAB240 antibody recognizes and binds to an epitope that becomes exposed upon the unfolding of the DBD. This manuscript shows by multiple experiments that the DBD of the β- and the ɣ-isoforms are not compromised but that the oligomerization domain is not functional. In figures 3D and 3E we have not included the TA β- and the ɣ-isoforms, because, again, they have a folded DBD and their inclusion would not provide any additional information compared to TAp53α.

(4) Figures 4B and 4C are small and extremely difficult to read.

We agree and have rearranged and enlarged these and other figures. Please see also answer to comment (5) of reviewer 1.

(5) Figure 5C: the authors claim that "the isoform induced cellular stress that triggers the expression of chaperones" (line 320). However, if the induction of the HSP70 promoter is shown, there is no evidence that this is due to cellular stress. Evidence to support that claim should be shown.

The expression and accumulation of unfolded, aggregation prone sequences is a stress situation for the cell which triggers the expression of chaperones. The expression of isoforms that are not well folded or of p53 mutants that are not well folded increases expression both from the HSP70 promoter and the heat shock promoter. This shows that the expression of unfolded isoforms induces cellular stress.

(6) Figure 5D: why was this experiment performed in SAOS2 cells when the whole paper was otherwise performed in H1299 cells?Also, about this figure, the authors write "In addition to this common set, Δ133p53α and Δ40p53α showed only very few additional interaction partners. This situation was very different for Δ133α, Δ133β and TAp53γ." (lines 331 to 333). My feeling is that we should instead read "In addition to this common set, TAp53β and Δ40p53α showed only very few additional interaction partners. This situation was very different for Δ133p53α, Δ133p53β and TAp53γ"

Thank you for spotting this mistake. Indeed, the correct wording is TAp53β and Δ40p53α and we have corrected the manuscript.

The mass spectrometry experiments were actually not carried out in SAOS cells, but in U2OS cells. The reason for not using the H1299 cell line was that these cells do not contain functional p53. In contrast, U2OS cells express wild type p53. We have repeated the mass spectrometry analysis and analyzed the data with a special focus on p53 peptides. This information is now added as Figure 5E. In this analysis we show that the Δ133p53α samples contain peptides from the DBD that are not part of this truncated isoform and must therefore originate from wild type p53 with which this isoform hetero-oligomerizes. The corresponding peptides are absent from Δ133p53β, showing that without a functional oligomerization domain this isoform does not interact with wild type p53. Likewise, the data demonstrate that the β- and the ɣ-isoforms do not form hetero-oligomers.

(7) Supplementary Table 2: the authors claim "For Δ133p53α we could identify peptides between amino acids 102 and 132 that must originate from wild type p53". SAOS2 has a WT TP53 gene and expresses all isoforms endogenously. Therefore, peptides between amino acids 102 and 132 can actually originate from "WT p53" but also TAp53β, TAp53γ, Δ40p53α, Δ40p53β or Δ40p53γ (most likely a mix of these).

We have not used SAOS cells but U2OS cells. As mentioned above the data show that the Δ133p53α sample contains peptides from wild type p53 and that these peptides cannot be found in the Δ133p53β sample. In addition, peptides originating from the oligomerization domain are only found in the samples of isoforms containing an oligomerization domain but not in samples of β- and ɣ-isoforms. The data are presented in Figure 5 E-G and Supplementary Figure S5.

Since the Biotin ligase is directly fused to a specific isoform, peptides from other isoforms can only be detected if these directly interact with the isoform fused to the ligase (and contain unique peptides, not present in the isoform fused to the ligase). The data confirm that only isoforms that have a functional oligomerization domain can interact with wild type p53 (or potentially other isoforms with a functional oligomerization domain).

(8) Figure 6: Why not conduct these luciferase reporter assays using the MDM-2 and p21 promoters like in Figure 2B and 2C since there may be promoter-specific regulation?This would be particularly important for the p21 promoter as TAp53β is known to induce it (Fujita K. et al. p53 isoforms Delta133p53 and p53beta are endogenous regulators of replicative cellular senescence. Nat Cell Biol. 2009 Sep;11(9):1135-42) and the Δ133p53α, Δ133p53β and Δ133p53γ isoforms were shown to reduce p21 transcription by TAp73β when co-expressed in H1299 cells (Zorić A. et al. Differential effects of diverse p53 isoforms on TAp73 transcriptional activity and apoptosis. Carcinogenesis. 2013 Mar;34(3):522-9.). Neither of these regulations appears here on the pBDS2 reporter, which is puzzling.

The main point of this paper is that all isoforms without a complete DNA binding domain and without a complete oligomerization domain do not bind to DNA with high affinity and do not show transcriptional activity and that is independent of the promotor. There might be effects of expressing certain isoforms in some cells, but that is most likely by inducing a stress response via expression of chaperones etc. High affinity sequence specific DNA binding does not play a role here (see results in Figure 2) and we have therefore not conducted these suggested experiments.